# Trypanosomatid selenophosphate synthetase structure, function and interaction with selenocysteine lyase

**Marco Túlio Alves da Silva**[1◉]**, Ivan Rosa e Silva**[1◉]**, Lívia Maria Faim**[1]**, Natália Karla Bellini**[1]**, Murilo Leão Pereira**[1]**, Ana Laura Lima**[1]**, Teresa Cristina Leandro de Jesus**[1,2]**, Fernanda Cristina Costa**[1,3]**, Tatiana Faria Watanabe**[4]**, Humberto D'Muniz Pereira**[1]**, Sandro Roberto Valentini**[4]**, Cleslei Fernando Zanelli**[4]**, Júlio Cesar Borges**[5]**, Marcio Vinicius Bertacine Dias**[6]**, Júlia Pinheiro Chagas da Cunha**[2]**, Bidyottam Mittra**[7]**, Norma W. Andrews**[7]**, Otavio Henrique Thiemann**[1,8]** *

**1** Laboratory of Structural Biology, Sao Carlos Institute of Physics, University of São Paulo, São Carlos, SP, Brazil, **2** Laboratory of Cell Cycle and Center of Toxins, Immune Response and Cell Signaling—CeTICS, Butantan Institute, São Paulo, SP, Brazil, **3** London School of Hygiene and Tropical Medicine, London, United Kingdom, **4** School of Pharmaceutical Sciences, São Paulo State University (UNESP), Araraquara, SP, Brazil, **5** São Carlos Institute of Chemistry, University of São Paulo, São Carlos, SP, Brazil, **6** Department of Microbiology, Institute of Biomedical Science, University of São Paulo, São Paulo, SP, Brazil, **7** Department of Cell Biology and Molecular Genetics, University of Maryland, College Park, Maryland, United States of America, **8** Department of Genetics and Evolution, Federal University of São Carlos, São Carlos, SP, Brazil

◉ These authors contributed equally to this work.
* thiemann@ifsc.usp.br

**Data Availability Statement:** All relevant data are within the manuscript and its Supporting Information files.

## Abstract

Eukaryotes from the Excavata superphylum have been used as models to study the evolution of cellular molecular processes. Strikingly, human parasites of the Trypanosomatidae family (*T. brucei*, *T. cruzi* and *L. major*) conserve the complex machinery responsible for selenocysteine biosynthesis and incorporation in selenoproteins (SELENOK/SelK, SELENOT/SelT and SELENOTryp/SelTryp), although these proteins do not seem to be essential for parasite viability under laboratory controlled conditions. Selenophosphate synthetase (SEPHS/SPS) plays an indispensable role in selenium metabolism, being responsible for catalyzing the formation of selenophosphate, the biological selenium donor for selenocysteine synthesis. We solved the crystal structure of the *L. major* selenophosphate synthetase and confirmed that its dimeric organization is functionally important throughout the domains of life. We also demonstrated its interaction with selenocysteine lyase (SCLY) and showed that it is not present in other stable assemblies involved in the selenocysteine pathway, namely the phosphoseryl-tRNA[Sec] kinase (PSTK)-Sec-tRNA[Sec] synthase (SEPSECS) complex and the tRNA[Sec]-specific elongation factor (eEFSec) complex. Endoplasmic reticulum stress with dithiothreitol (DTT) or tunicamycin upon selenophosphate synthetase ablation in procyclic *T. brucei* cells led to a growth defect. On the other hand, only DTT presented a negative effect in bloodstream *T. brucei* expressing selenophosphate synthetase-RNAi. Furthermore, selenoprotein T (SELENOT) was dispensable for both forms of the parasite. Together, our data suggest a role for the *T. brucei* selenophosphate synthetase in the regulation of the parasite's ER stress response.

**Funding:** MTAS (FAPESP 11/24017-4 and 13/
02848-7), IRS (FAPESP 10/04429-3), LMF
(FAPESP 07/06591-0), FCC (FAPESP 08/58501-7),
TCLJ (FAPESP 11/06087-5), OHT (FAPESP 06/
55685-4, 08/57910-0). NKB, ALL and MLP are
thankful for CAPES and CNPq institutional
scholarships. FAPESP: Fundação de amparo a
pesquisa do estado de São Paulo (www.fapesp.br)
CAPES: Coordenação de Aperfeiçoamento de
Pessoal de Nível Superior (https://www.capes.gov.
br) CNPq: Conselho Nacional de Desenvolvimento
Científico e Tecnológico (www.cnpq.br) The
funders had no role in study design, data collection
and analysis, decision to publish, or preparation of
the manuscript.

**Competing interests:** The authors have declared
that no competing interests exist.

## Author summary

Selenium is both a toxic compound and a micronutrient. As a micronutrient, it partici-
pates in the synthesis of specific proteins, selenoproteins, as the amino acid selenocysteine.
The synthesis of selenocysteine is present in organisms ranging from bacteria to humans.
The protist parasites of the Trypanosomatidae family, that cause major tropical diseases,
conserve the complex machinery responsible for selenocysteine biosynthesis and incorpo-
ration in selenoproteins. However, this pathway has been considered dispensable for the
parasitic protist cells. This has intrigued us, and lead to question that if maintained in the
cell it should be under selective pressure and therefore be necessary. Also, extensive and
dynamic protein-protein interactions must happen to deliver selenium-containing inter-
mediates along the pathway in order to warrant efficient usage of biological selenium in
the cell. In this study we have investigated the molecular interactions of different proteins
involved in selenocysteine synthesis and its putative involvement in the endoplasmic retic-
ulum redox homeostasis.

## Introduction

*Trypanosoma brucei*, *Trypanosoma cruzi* and *Leishmania sp*. protist parasites [1] are collec-
tively responsible for thousands of productive life years lost worldwide, as a consequence of
human sleeping sickness [2], Chagas' disease [3], and leishmaniasis [4], respectively. They
cycle between an insect vector and a mammalian host, progressing through different life-cycle
stages with varying metabolism, cell morphology and surface architecture. Adverse environ-
mental conditions such as nutrient deficiency, hypoxia, oxidative stress, pH and temperature
variation occur throughout their life cycle [5]. Trypanosomatids depend on dynamic gene
expression to regulate their adaptation to stress, differentiation and proliferation, in response
to diverse environmental signals within different hosts [5,6]. Interestingly, gene expression is
controlled post-transcriptionally by spliced leader (SL) trans-splicing, RNA editing and
mRNA stability [6].

  In their life cycle, these parasites are exposed to reactive oxygen species that are controlled
by a unique thiol-redox system based on trypanothione reductase, tryparedoxin and trypare-
doxin peroxidase [7,8,9]. In contrast, the main redox regulatory enzymes in mammals are
thioredoxin and gluthatione reductases, which contain the L-selenocysteine (Sec) residue in
the active site [10]. Remarkably, only three selenoproteins, namely SELENOK, SELENOT and
SELENOTryp, have been reported in trypanosomatids to contain Sec-based putative redox
centers, as confirmed by $^{75}$Se-labeled homologs from *T. brucei* [11], *T. cruzi* [12] and *L. dono-
vani* [13]. Selenocysteine biosynthesis and incorporation into selenoproteins require an intri-
cate molecular machinery that is present, but not ubiquitous, in all domains of life [14]. In
eukaryotes [14,15,16] it begins with tRNA$^{[Ser]Sec}$ acylation with L-serine by the seryl-tRNA
synthetase (SerRS) followed by its conversion to Sec-tRNA$^{[Ser]Sec}$, sequentially catalyzed by
phosphoseryl-tRNA$^{Sec}$ kinase (PSTK) and Sec-tRNA$^{[Ser]Sec}$ synthase (SEPSECS). Selenopho-
sphate synthetase (SEPHS) is a key enzyme in the Sec pathway, being responsible for catalyzing
the formation of the active selenium donor for this reaction, selenophosphate, from selenide
and ATP. Finally, a tRNA$^{Sec}$-specific elongation factor (eEFSec) directs the Sec-tRNA$^{[Ser]Sec}$
molecule to the ribosome in response to an UGA$_{Sec}$ codon, in the presence of a Sec insertion
sequence (SECIS) in the mRNA. However, details of the protein-protein and protein-RNA

interaction network and the mechanism of selenoprotein biosynthesis and its regulation in trypanosomatids remain poorly understood [12,17].

Furthermore, little is known about selenium metabolism itself in trypanosomatids [17]. It has been reported that selenium in trace amounts is essential for several organisms throughout several domains of life, although at high concentration it is cytotoxic [17,18]. Selenocysteine is specifically decomposed by selenocysteine lyase (SCLY) into L-alanine and selenide, which is potentially reused by selenophosphate synthetase in several eukaryotes, including trypanosomatids [19,20,21]. Mammals conserve two paralogues of selenophosphate synthetase, namely SEPHS1 and SEPHS2. Mammalian SEPHS2 is itself a selenoprotein known to be essential for selenophosphate formation and consequently selenoprotein biosynthesis. In contrast, the SPS1 isoform (SEPHS1) does not conserve a cysteine or selenocysteine residue in the catalytic site and is likely involved in redox homeostasis regulation [22,23,24]. Strikingly, the cellular availability of hydrogen peroxide is altered by SEPHS1 deficiency in embryonic mammalian cells [25]. Our group previously showed that trypanosomatids conserve only one selenophosphate synthetase, SEPHS2 (SPS2), containing a catalytic cysteine [26]. Its function has been related to selenoprotein synthesis [11].

Not only T. brucei SEPHS2 (TbSEPHS2) but also PSTK, SEPSECS and eEFSec (TbPSTK, TbSEPSECS and TbeEFSec, respectively) [11] independent knockdowns impair selenoprotein synthesis in the parasite procyclic form (PCF). Interestingly, TbPSTK and TbSEPSECS double-knockout cell lines demonstrated that T. brucei PCF does not depend on selenoproteins [11]. This result was extended with the observation that TbSEPSECS is not essential for normal growth of the bloodstream form (BSF) T. brucei [27] and its survival in the mammalian host [28]. In addition, a TbSEPSECS knockout cell line did not show any growth disruption upon hydrogen peroxide-induced oxidative stress in PCF [27]. This result also seems to be valid for other trypanosomatids, since L. donovani SEPSECS null mutant promastigote (LdSEPSECS) cell lines show normal growth even upon oxidative stress, and during macrophage infection [13]. On the other hand, TbSEPHS2 RNAi PCF and BSF T. brucei cells are sensitive to oxidative stress induced by hydrogen peroxide [29]. The reason why trypanosomatids maintain such complex machinery for selenoprotein biosynthesis remains unclear.

Moreover, the function of Kinetoplastea selenoproteins has not been elucidated yet. Interestingly, T. brucei PCF and BSF are sensitive to nanomolar concentrations of auranofin [12,17], an inhibitor of mammalian selenoprotein biosynthesis. Nonetheless, auranofin did not show any differential effect on TbSEPSECS knockout lines, when compared to the wild type T. brucei PCF [27] or the counterpart L. donovani amastigote [13]. SELENOTryp (Sel-Tryp) is a novel selenoprotein exclusive to trypanosomatids that contains a conserved C-terminal redox motif, often found in selenoproteins that carry out redox reactions through the reversible formation of a selenenylsulfide bond [12]. On the other hand, mammalian SELENOK/SelK [30] and SELENOT/SelT [31] homologs were recently shown to be endoplasmic reticulum (ER) residents, where they have a role in regulating $Ca^{2+}$ homeostasis. Regulation of ER redox circuits control homeostasis and survival of cells with intense metabolic activity [30,31]. Chemical induction of ER stress with DTT and tunicamycin in PCF T. brucei, but not BSF [32] apparently results in ER expansion and elevation in the ER chaperone BiP, inducing the unfolded protein response (UPR) [33,34,35]. Prolonged ER stress induces the spliced leader RNA silencing (SLS) pathway [34]. Induction of SLS, either by prolonged ER stress or silencing of the genes associated with the ER membrane that function in ER protein translocation, lead to programmed cell death (PCD). This result is evident by the surface exposure of phosphatidyl serine, DNA laddering, increase in ROS production and cytoplasmic $Ca^{2+}$, and decrease of mitochondrial membrane potential [34].

Despite the wealth of information on the selenocysteine machinery in eukaryotes, seleno-protein biosynthesis and function in the superphylum Excavata remain poorly understood. Here, we present a detailed structural, biochemical and functional analysis of trypanosomatid selenophosphate synthetase (*Tb*SEPHS2). The *Tb*SEPHS2 crystal structure demonstrates that a conserved aminoimidazole ribonucleotide synthetase (AIRS)-like fold is important for its function. We also show that *Tb*SEPHS2 interacts with *T. brucei* selenocysteine lyase (*Tb*SCLY) *in vitro* and that they co-purify from procyclic *T. brucei* cell extracts. We further demonstrate that the *Tb*SEPHS2-SCLY binary complex is not part of other stable complexes in the Sec-pathway of *T. brucei*, namely the *Tb*SEPSECS-tRNA[Ser]Sec-PSTK complex and the *Tb*EFSec-tRNA[Ser]Sec complex. We also determined that *Tb*SEPHS2 ablation in procyclic *T. brucei* cells leads to growth defect in the presence of the ER stressors DTT or tunicamycin while only DTT showed a negative effect in bloodstream cells. Also, SELENOT was found to be dispensable for both PCF and BSF *T. brucei*. Together, our data shed light into the protein assemblies involved in the selenocysteine pathway in *T. brucei* and suggest a possible role for the *T. brucei* seleno-phosphate synthetase in regulation of the parasite's ER stress response.

## Results

### The *L. major* selenophosphate synthetase crystal structure is highly similar to its orthologs, despite sharing low amino acid sequence identity

*T. brucei* and *L. major* selenophosphate synthetases SEPHS2 isoforms have low sequence iden-tity to the well characterized orthologs from *Homo sapiens*, *Aquifex aeolicus* and *Escherichia coli* (42%, 29% and 28%, respectively) (S1 Fig). We described the crystallization of *Tb*SEPHS2 and ΔN(69)-*Lm*SEPHS2 elsewhere [36]. The full-length *Tb*SEPHS2 structure determination was not successful due to lack of sufficient experimental data, and the full-length *Lm*SEPHS2 was recalcitrant to crystallization. Here we present the crystal structure of ΔN-*Lm*SEPHS2 (PDB 5L16) at 1.9 Å resolution solved by molecular replacement using human SEPHS1 (PDB 3FD5) as a search model. The structure was refined to $R_{free}/R_{work}$ of 0.21/0.17 (detailed refine-ment statistics are shown in Table 1).

ΔN-*Lm*SEPHS2 crystallized as a monomer in the asymmetric unit showing a typical ami-noimidazole ribonucleotide synthetase (AIRS)-like fold [37], which consists of two α+β domains labeled N- and C-terminal AIRS (AIRS and AIRS_C, respectively) ranging from amino acid residues 74 to 190 and 204 to 384, respectively. The N-terminal AIRS domain folds into a six-stranded β-sheet flanked by two α-helices and one $3_{10}$-helix, while the AIRS_C domain also presents a six-stranded β-sheet that is flanked by seven α-helices and one $3_{10}$-helix (Fig 1A). Overall, the ΔN-*Lm*SEPHS2 monomer is highly similar to its orthologs, as revealed by the root mean square (R.M.S.D.) deviation of main-chain atomic positions between 0.7 Å and 2.2 Å when ΔN-*Lm*SEPHS2 is compared to *H. sapiens* SEPHS1 [22] (0.7 Å and 0.8 Å for PDBs 3FD5 and 3FD6, respectively), *A. aeolicus* SEPHS [38] (1.0 Å for PDBs 2ZAU, 2ZOD and 2YYE) and *E. coli* SEPHS (SelD) [39] (2.2 Å for PDB 3UO0) monomers, respectively (Fig 1B). The main differences occur in loops that are longer in *Lm*SEPHS. Our finding that the AIRS fold of selenophosphate synthetase is also conserved in our *L. major* crystal structure suggests its necessity for the enzyme mechanism.

Notably, selenophosphate synthetases were reportedly active dimers in *E. coli* [39], *A. aeoli-cus* [38] and *H. sapiens* [22]. Indeed, both recombinant full-length *Lm*SEPHS2 and *Tb*SEPHS2 predominantly oligomerize as elongated 84±3 kDa dimers *in vitro* as shown by native gel elec-trophoresis (Fig 1D and S2 Fig) and sedimentation velocity analytical ultracentrifugation (SV-AUC) (Fig 1E and 1F). Curiously, a relatively small amount of tetramers was also detected *in vitro* (Fig 1D–1F). Tetramer-dimer dissociation constants of 161±10 µM and 178±10 µM

**Table 1. ΔN-*Lm*SEPHS2 crystal structure refinement statistics.**

| Refinement | ΔN-*Lm*SEPHS2 |
|---|---|
| PDB code | 5L16 |
| Refinement program | REFMAC 5.8.0135 |
| Total number of atoms | 2,736 |
| Number of amino acid residues | 323 |
| Number of solvent atoms | 293 |
| Ligand | 1 molecule of sulfate ion |
| Resolution range (Å) (completeness) | 1.882–40.845 (96.1%) |
| Reflections used in refinement (in cross validation, random) | 33,411 (5%) |
| $R_{work}/R_{free}$ | 0.1732/0.2131 |
| Fo, Fc correlation | 0.95 |
| **B-factors (Å$^2$)** | |
| All atoms | 27.3 |
| Protein atoms | 18.0 |
| Ligand atoms | 46.5 |
| Water | 52.5 |
| **R.M.S.D** | |
| Bond lengths (Å) | 0.006 |
| Bond angles (°) | 1.018 |
| **Ramachandran plot (%)** | |
| Favored regions | 98.8 |
| Allowed regions | 99.7 |
| Outliers | 0.3 |
| **MolProbity** | |
| Clashscore | 2.46 |
| MolProbity score | 1.31 |

were measured by sedimentation equilibrium AUC (SE-AUC, S3 Fig) for *Tb*SEPHS2 and *Lm*SEPHS2, respectively. The data indicate that the dimer corresponds to the likely dominant form of selenophosphate synthetase in solution. An elongated dimer model of ΔN-*Lm*SEPHS2 (Fig 1C) was generated using PDBePISA [40] as a likely quaternary structure, stable in solution with a 3230 Å$^2$ buried surface. The dimerization surface occurs mainly between the β2 and β5 strands of adjacent monomers and is also stabilized by hydrophobic interactions between side chains, leading to the formation of an eight-stranded β-barrel. The dimeric structure of ΔN-*Lm*SEPHS2 conserves two symmetrically arranged ATP-binding sites, formed along the interface between AIRS and AIRS-C domains in each monomer. Amino acid residues previously described to bind ATP phosphate groups [22,38,39] are also conserved (Lys46, Asp64, Thr95, Asp97, Asp120, Glu173 and Asp279), although Lys46 and Asp64 are not present in the crystal structure (Fig 1 and S1 Fig). Interestingly, a novel sulfate binding site was identified in the ΔN-*Lm*SEPHS2 monomer at His84 and Thr85 (represented as sticks in Fig 1).

The N-terminal portion of selenophosphate synthetases has been shown to be highly flexible in the absence of ligand [22,38,39,41,42], and it is disordered in the crystal structure of the apo *A. aeolicus* SEPHS [42]. Similarly, the apo ΔN-*Lm*SEPHS2 crystal structure lacks a 69-amino acid residues-long N-terminal region that includes a glycine-rich loop, where the conserved catalytic residues Cys46 and Lys49 are located. A molecular tunnel formed by the long N-terminal loop in substrate-bound selenophosphate synthetase structures is believed to protect unstable catalysis intermediates [22,38,39]. Furthermore, the disordered SPSH2 N-

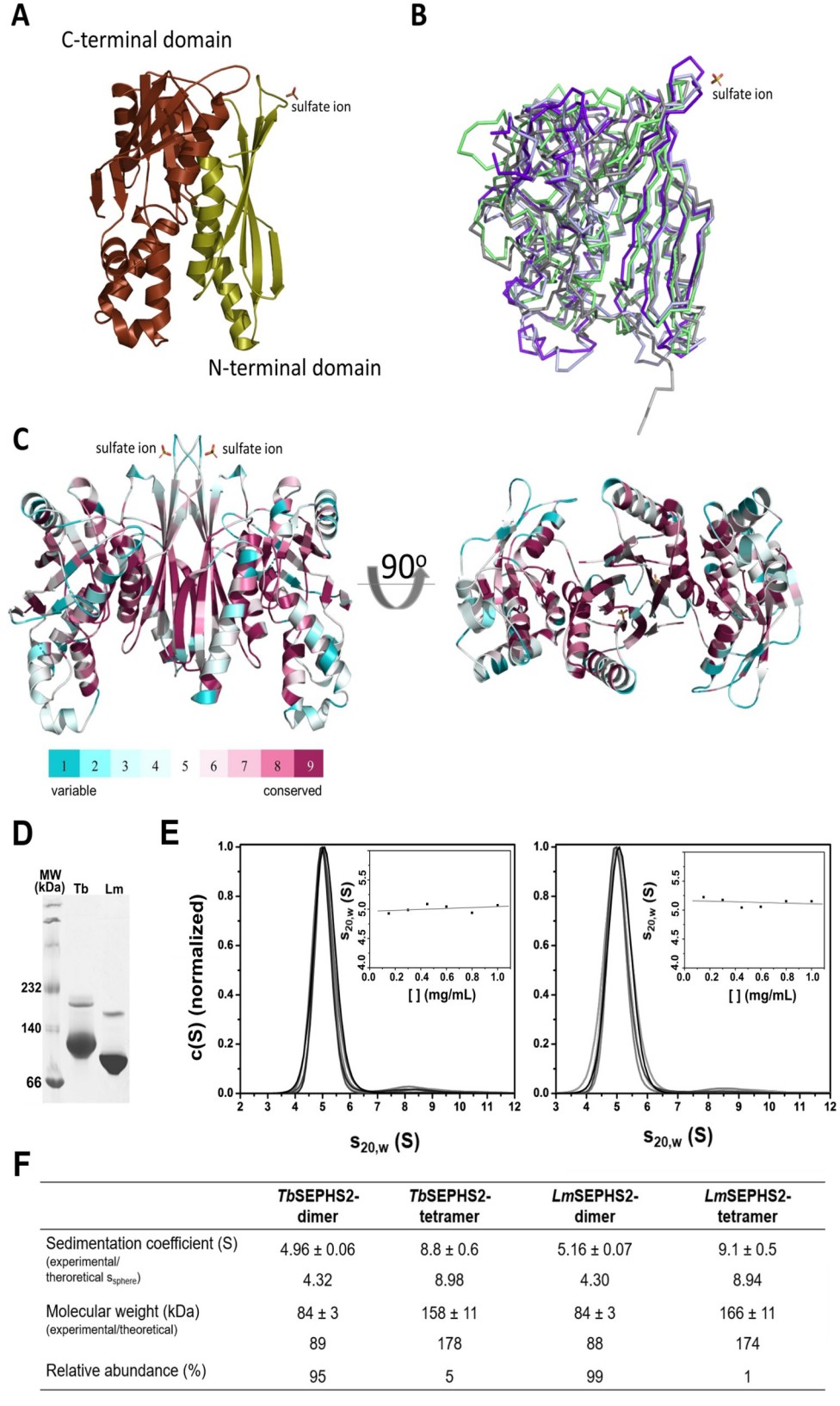

**Fig 1. ΔN-*Lm*SEPHS2 crystal structure. A-** Cartoon representation of the monomer structure in the asymmetric unit showing a typical AIRS-like folding. The sulfate ion is represented as sticks. **B-** Superimposition of *Aa*SEPHS (grey) [38], *Ec*SEPHS (green) [39], *Hs*SEPHS1 (light blue) [22] and *Lm*SEPHS2 (purple). **C-** Dimeric model generated using PDBePISA [40] depicting amino acid residue conservation. **D-** Native gel electrophoresis showing the prevalence of

dimers in solution for *T. brucei* (Tb) and *L. major* (Lm) selenophosphate synthetases at 2 mg/mL. A small amount of tetramers is also observed for both protein preparations (top bands). MW: molecular weight. **E**- Sedimentation coefficient distribution (S) at increasing protein concentration normalized to the most abundant oligomer (dimer) obtained by sedimentation velocity analytical ultracentrifugation (SV-AUC). The inset displays sedimentation coefficients measured for dimers at increasing total protein concentration. **F**- Measured and theoretical sedimentation coefficient, molecular weight and relative abundance of dimers and tetramers. $s_{sphere}$ corresponds to the theoretical sedimentation coefficient calculated for a spherical protein. The discrepancy between the experimental sedimentation coefficient for the dimer and its theoretical value ($s_{sphere}$) suggests that it is elongated.

terminal region does not seem to be essential for the protein dimerization in-vitro (S2 Fig) and its absence does not destabilize its secondary structure (S4A Fig).

### The N-terminal region of trypanosomatid SEPHS2 is important but not essential for ATPase activity *in vitro* and selenoprotein biosynthesis in *selD*-deficient *E. coli*

Our group previously showed that both *Tb*SEPHS2 and *Lm*SEPHS2 have a slow kinetics *in vitro* in the presence of selenide [26]. We further evaluated their ATPase activity in the absence of selenide by monitoring the ATP peak over time by HPLC, as shown in Fig 2A. Full-length *Lm*SEPHS2 consumed most of the ATP available *in vitro* during the first five hours of reaction, while full-length *Tb*SEPHS2 consumed half of it during the same period. Interestingly, ΔN(25)-*Tb*SEPHS2, which lacks the predicted disordered N-terminus but preserves all catalytic residues, consumed half of the available ATP only after an 18 hour reaction, indicating that this region is important but not essential for its ATPase activity. On the other hand, ΔN(70)-*Tb*SEPHS2 constructs, which lack the functional residues necessary for selenophosphate formation but conserve most amino acid residues composing the two ATP-binding sites, showed only small residual ATP hydrolysis in the absence of selenide. Curiously, ΔN-*Lm*SEPHS2 showed residual ATPase activity comparable to ΔN(25)-*Tb*SEPHS2. As a control, ATP did not show any residual hydrolysis in the absence of selenophosphate synthetase even after 72 hours of incubation in the reaction buffer.

Like *E. coli* SEPHS (SelD) [39], but in contrast with its *H. sapiens* orthologs, trypanosomatid SEPHS2 is not a selenoprotein itself [26]. *Lm*SEPHS2 was previously shown to restore selenoprotein biosynthesis in a SEPHS deficient *E. coli* WL400(DE3) strain [26]. To extend this information, we verified that both full-length *Tb*SEPHS2 and ΔN(25)-*Tb* SEPHS2, but not ΔN(70)-*Tb*SEPHS2 and ΔN-*Lm*SEPHS2, are also capable of complementing *selD* deletion in *E. coli* (Fig 2B). As expected, no functional complementation resulted from Cys42Ala-*Tb*SEPHS2 and Cys46Ala-*Lm*SEPHS2 mutants, as negative controls (Fig 2B). Notably, although having slow kinetics *in vitro*, ΔN(25)-*Tb*SEPHS2 successfully restored *E. coli* SEPHS function (Fig 2B).

### *T. brucei* SEPHS2 binds selenocysteine lyase (*Tb*SCLY) but does not co-purify with higher order complexes of the selenocysteine pathway from *T. brucei*

A putative interaction of eukaryotic selenophosphate synthetase with selenocysteine lyase has been suggested based on reported co-immunoprecipitation of mouse homologs [43]. Thus, we sought to evaluate the *T. brucei* SCLY-SEPHS2 direct interaction *in vitro* by SEC-MALS. We unambiguously observed a binary hetero-complex formation *in vitro* (Fig 3A). Isothermal titration calorimetry (ITC) confirmed the interaction (Fig 3B and S5A Fig). We also determined that the pyridoxal-phosphate (PLP) molecule bound to the active sites of SCLY is hidden upon SEPHS2 interaction, as measured by a decrease in PLP fluorescence accompanied by a blue shift (Fig 3C). Additionally, we observed that ΔN(70)-*Tb*SEPHS2 does not bind *Tb*SCLY as measured by ITC (S5B Fig), indicating that the N-terminal region is necessary for *in vitro*

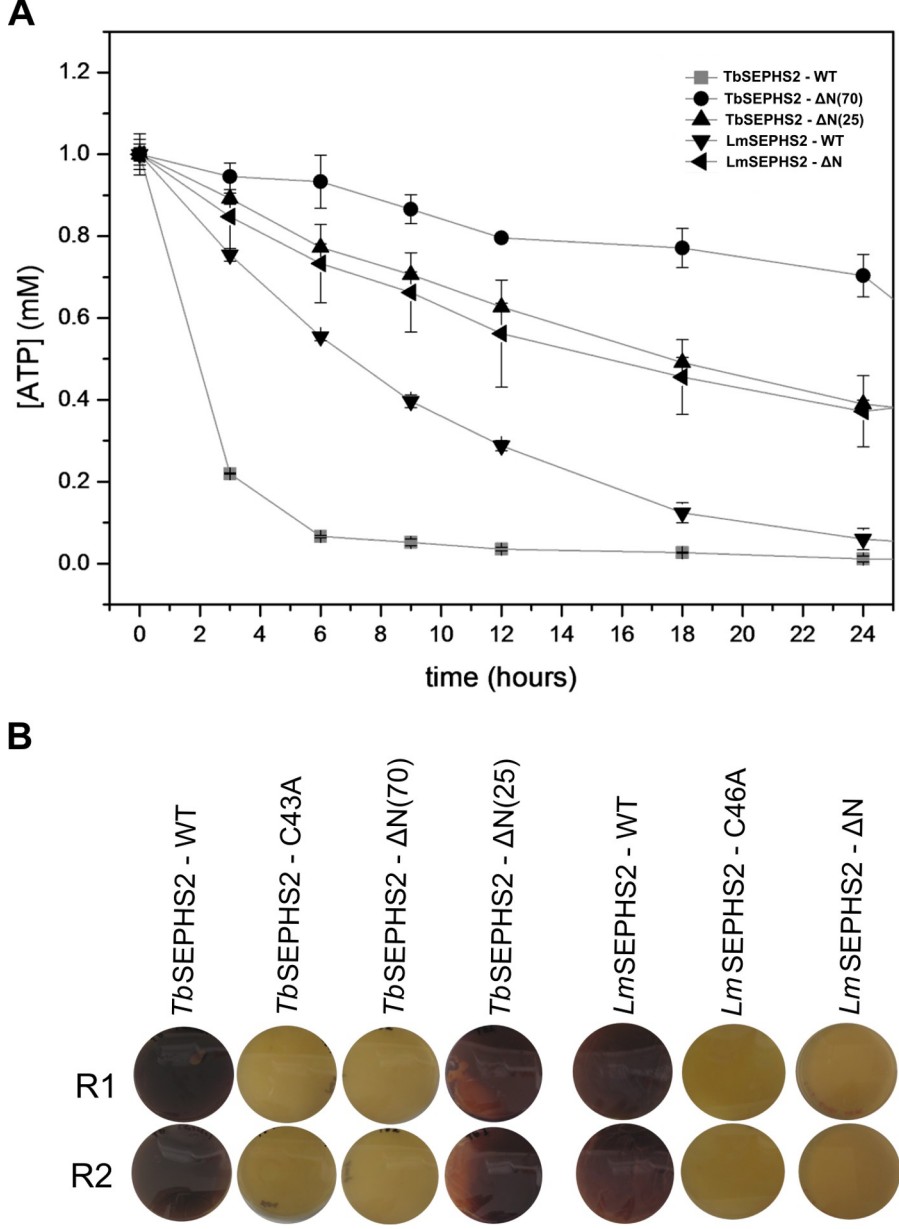

**Fig 2. ATPase activity and functional complementation assays. A-** ATP hydrolysis *in vitro* over time measured by HPLC for full length and N-terminally truncated constructs of *T. brucei* and *L. major* selenophosphate synthetases. **B-** Selenophosphate synthetase functional complementation assays in SEPHS deficient *E. coli* strain (WL400 (DE3)) transformed with different constructs. The purple color indicates a functional formate dehydrogenase H selenoprotein expression. R1 and R2 correspond to biological duplicates.

interaction, similarly to the *E. coli* SEPHS (*Ec*SEPHS) N-terminal dependence for SEPHS-SelA-tRNA[Ser]Sec ternary complex formation in *E. coli* [41].

Importantly, *Tb*SEPHS2 and *Tb*SCLY co-purified with each other in independent PTP (protein A—TEV site—protein C)-TAP (tandem affinity purification) experiments (Fig 3D and 3F, and S1 Table). However, no tRNA[Ser]Sec was copurified in either experiment (Fig 3E), suggesting that this interaction may occur independent of tRNA[Ser]Sec. Interestingly, *Tb*SCLY was previously reported to localize predominantly to the nucleus of PCF *T. brucei* [21]. On the

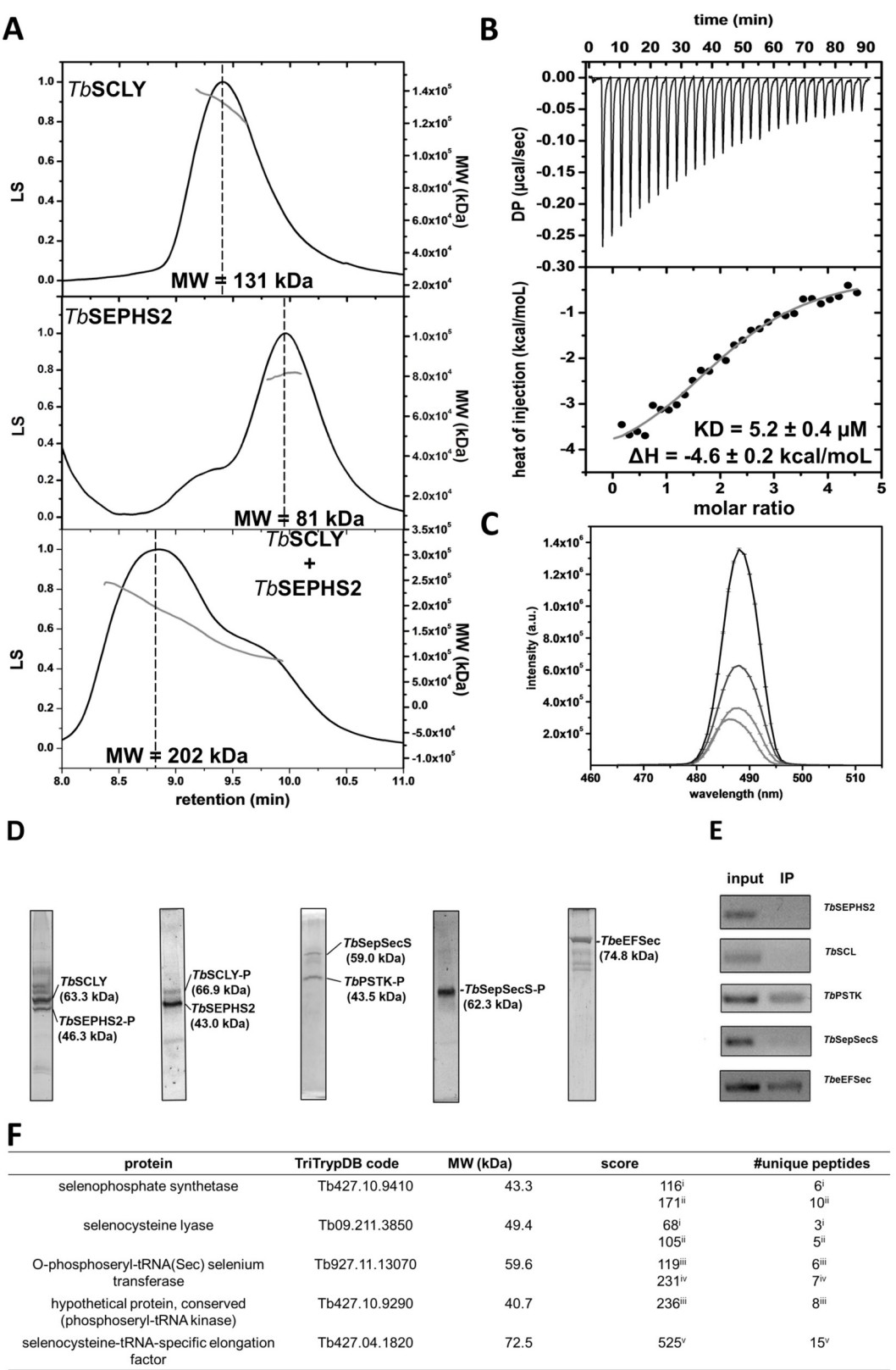

**Fig 3. Interaction between selenophosphate synthetase and selenocysteine lyase. A-** SEC-MALS profiles for *Tb*SCLY
(40 μM; theoretical molecular weight (dimer) = 127 kDa), *Tb*SEPHS2 (40 μM; theoretical molecular weight (dimer) = 89

| protein | TriTrypDB code | MW (kDa) | score | #unique peptides |
|---|---|---|---|---|
| selenophosphate synthetase | Tb427.10.9410 | 43.3 | 116[i] 171[ii] | 6[i] 10[ii] |
| selenocysteine lyase | Tb09.211.3850 | 49.4 | 68[i] 105[ii] | 3[i] 5[ii] |
| O-phosphoseryl-tRNA(Sec) selenium transferase | Tb927.11.13070 | 59.6 | 119[iii] 231[iv] | 6[iii] 7[iv] |
| hypothetical protein, conserved (phosphoseryl-tRNA kinase) | Tb427.10.9290 | 40.7 | 236[iii] | 8[iii] |
| selenocysteine-tRNA-specific elongation factor | Tb427.04.1820 | 72.5 | 525[v] | 15[v] |

[i]*Tb*SEPHS2-PTP, [ii]*Tb*SCLY-PTP, [iii]TbPSTK-PTP, [iv]*TbSepSecS*-PTP, [v]*Tb*eEFSec-PTP

kDa) and 1 *Tb*SCLY (40 μM): 1 *Tb*SEPHS (40 μM) indicating the formation of a binary complex *in vitro*. The molecular weight (MW) corresponding to the highest peak centroid is indicated. **B**- ITC curves obtained by *in vitro* titration of *Tb*SEPHS2 (200 μM, syringe) to *Tb*SCLY (10 nM, calorimeter cell). **C**- *Tb*SCLY-PLP (20 μM) fluorescence in the presence of different concentrations of *Tb*SEPHS2 (1:0.5, 1:0.75, 1:1, 1:1.25, 1:1.5). **D**- SyproRuby™ stained SDS-PAGE of tandem affinity purification (TAP) products of either *Tb*SEPHS2-PTP, *Tb*SCLY-PTP, *Tb*PSTK-PTP, *Tb*SEPSECS-PTP or *Tb*eEFSec-PTP as baits. **E**- Analysis of tRNA^Sec copurification by RT-PCR. Input: lysate expressing the respective PTP-tagged protein. IP: Immunoprecipitated complex using anti-IgG beads. **F**- LC-MS/MS analysis of the corresponding SDS-PAGE bands.

other hand, we immunolocalized a C-terminally PTP-tagged construct of *Tb*SEPHS2 both in the nucleus and the cytoplasm of the cell (S6 Fig). We further observed that *Tb*PSTK sub-cellular localization in PCF *T. brucei* is similar to *Tb*SPSH2, whereas *Tb*SEPSECS and *Tb*eEFSec were excluded from the nucleus (S6 Fig). Since a colocalization experiment was not possible due to the unavailability of antibodies against each protein, we sought to evaluate the formation of putative larger complexes involved in the Sec pathway (Fig 4A) using PTP-TAP experiments of *Tb*PSTK-PTP, *Tb*SEPSECS-PTP and *Tb*eEFSec-PTP. *Tb*SEPSECS-PTP did not co-purify any other protein, whereas *Tb*PSTK-PTP co-purified with *Tb*SEPSECS (Fig 3D). The C-terminal PTP-tag of *Tb*SEPSECS might have impeded its interaction with *Tb*PSTK. Additionally, no stable protein-protein complex was observed for *Tb*eEFSec-PTP (Figs 3D and 4A). RT-PCR analysis revealed that tRNA^[Ser]Sec co-precipitates with the *Tb*PSTK-P-SEPSECS complex and with *Tb*eEFSec (Fig 3E).

Furthermore, poly-ribosomal profiling experiments showed that neither *Tb*SEPHS2 nor *Tb*SCLY are present in ribosomal complexes involved in the Sec pathway in PCF *T. brucei* (Fig 4B and 4C, respectively). Also, *Tb*PSTK and *Tb*SEPSECS were mostly present in ribosome-free fractions (Fig 4B and 4D, respectively). A small amount of these proteins was detected in monosome fractions possibly due to an overlap between ribosome-free and 40S ribosome fractions, as confirmed by the detection of BiP control in both fractions (Fig 4) consistent with Small-Howard et al [44] results that showed that *Hs*SEPHS1 is also present in mammalian in ribosome-free fractions. Additionally, *Tb*eEFSec was detected in all the fractions, including present in 80S ribosomes and polysomes (Fig 4E), as shown for *Hs*eEFSec [44]. Disruption of the mounted ribosome with the chelating agent EDTA demonstrated that *Tb*eEFSec dissociated from monosomes or polysomes, being detected only in mRNA-free fractions (Fig 4F).

## *Tb*SEPHS2 RNAi-induced *T. brucei* cells are sensitive to endoplasmic reticulum chemical stressors

Ablation of selenophosphate synthetase function impairs selenoprotein synthesis not only in mammals [24] but also in *T. brucei* [11]. However, *Tb*SEPHS2 is not essential for either PCF and BSF *T. brucei* under laboratory-controlled conditions [11,27]. Recent work showed that mammalian selenoprotein T (SELENOT) is involved in ER stress response [31]. Therefore, we sought to investigate whether chemical ER stress upon SEPHS2 ablation impairs *T. brucei* viability. We induced RNAi expression against *Tb*SEPHS2 with tetracycline for 48 hours and the cells were subsequently treated with common stressors of ER, namely DTT and tunicamycin (TN) for two hours.

Both DTT and TN caused a slight but significant reduction in viability of induced PCF *T. brucei* cells, suggesting that they negatively interfere with ER metabolism in the absence of *Tb*SEPHS2 (Fig 5A and 5C). Additionally, *Tb*SEPHS2-RNAi BSF *T. brucei* cells were induced with tetracycline for 24 hours and subsequently incubated with TN or DTT for two hours. Treatment with DTT at 150 and 300 μM showed a reduction of *Tb*SEPHS2-RNAi BSF *T. brucei* cells (Fig 5B) viability. No effect was detected at any concentration of TN in BSF *T. brucei*

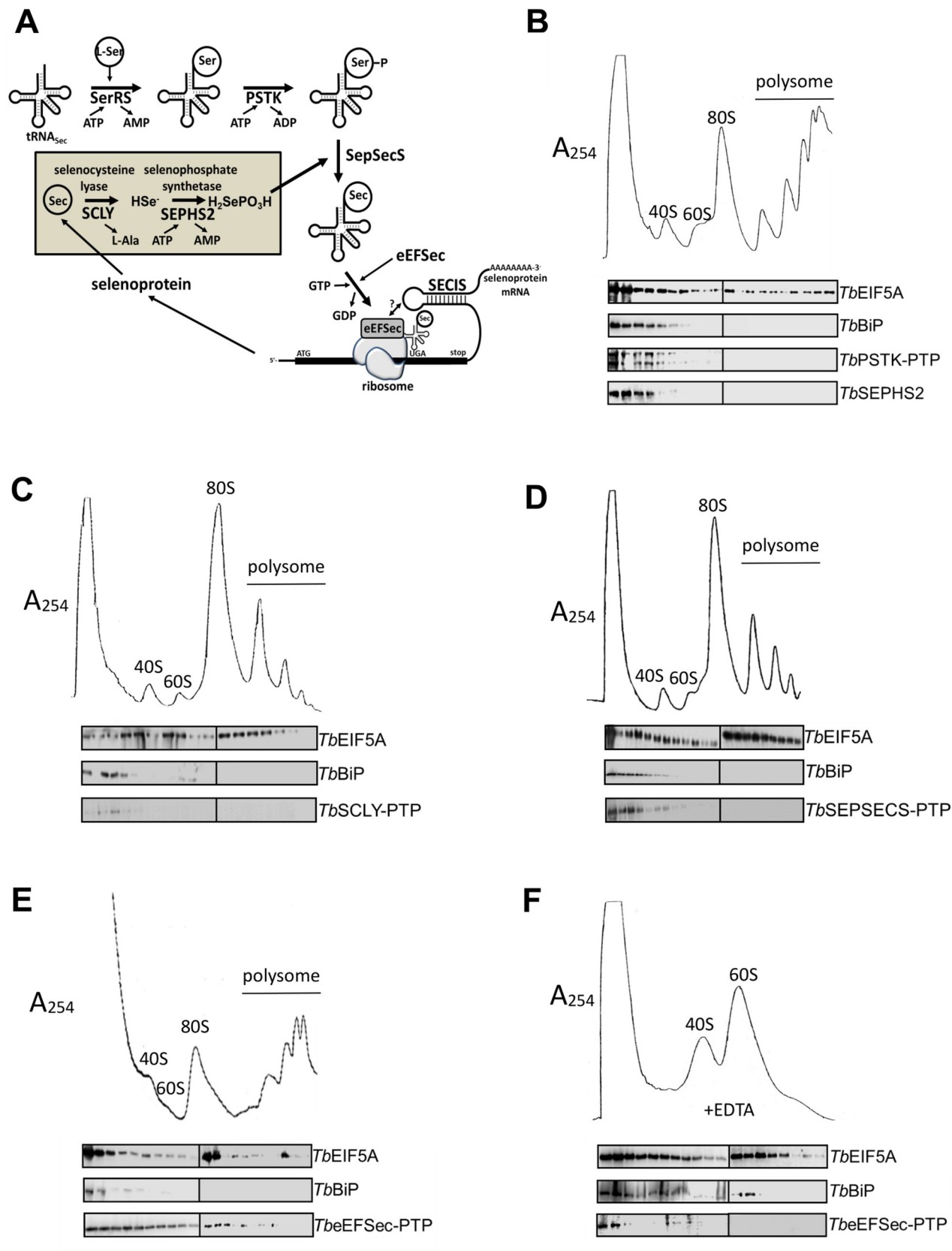

**Fig 4. Polysomal profile analysis of selenoprotein synthesis factors. A-** Schematic representation of the selenocysteine pathway in trypanosomatids. Lysates of PCF *T. brucei* PTP-tagged selenocysteine biosynthesis proteins were fractionated in a sucrose gradient centrifugation (7–47% sucrose) as ribosome-free, monosome (40S, 60S and 80S) and polysome fractions as monitored by UV absorbance at 254 nm. Western blot analyses of tagged proteins, using anti-protein A antibody were carried out to localize selenoprotein synthesis factors (**B-** *Tb*PSTK-PTP, **C-** *Tb*SCLY-PTP, **D-** *Tb*SEPSECS-PTP, and **E-** *Tb*eEFSec-PTP). BiP and EIF5A were used as ribosome-free and polysome fraction markers, respectively. **F-** Ribosomes dissociation into monosome units in the presence of EDTA fractionated in a 5–25% sucrose gradient.

(Fig 5D). Also, no alteration of BiP expression in both *Tb*SEPHS2-RNAi PCF and BSF *T. brucei* cells was observed by Western blot (Fig 5I and 5J, respectively).

## Selenoprotein T (SELENOT) is not essential for both procyclic and bloodstream forms of *T. brucei*

The mammalian selenoprotein T (SELT, SELENOT) is an ER-resident enzyme whose Sec-containing redox domain is believed to regulate various post-translational modifications that require protein disulfide bond formation in the ER including chaperones and also contributing to $Ca^{2+}$ homeostasis [31]. Thus, we sought to evaluate whether this enzyme is essential in *T. brucei*. Tetracycline-induced *Tb*SELENOT-RNAi resulted in 96% reduction of *Tb*SELENOT mRNA in PCF *T. brucei* as measured by qPCR, with no significant growth defect compared to non-induced cells (Fig 6A and 6C). In BSF *T. brucei*, a slight growth defect was observed for tetracycline-induced cells (Fig 6B and 6D) with around 91% mRNA level reduction.

We further evaluated the SELENOT response to ER stress upon DTT and TN treatment. Interestingly, *Tb*SELENOT-RNAi-induced PCF cells were only sensitive to intermediate concentrations of DTT (1.0 nM) and TN (10 μM) (Fig 7A and 7C, respectively). On the other hand, no significant concentration-dependent effect was observed for *Tb*SELENOT-RNAi-BSF *T. brucei* in the presence of DTT or TN. (Fig 7B and 7D, respectively). Moreover, stable tetracycline-induced *Tb*SELENOLT-RNAi cell lines did not show any increase in BiP expression in either PCF or BSF *T. brucei* (Fig 7E and 7F, respectively).

Additionally, we sought to test if lack of SELENOT alters the sensitivity of *T. brucei* to hydrogen peroxide. Treatment with different concentrations of hydrogen peroxide did not affect the growth of any of the *T. brucei* forms upon SELENOT depletion (Fig 8A and 8B), ruling out a putative SELENOT role in oxidative stress protection in *T. brucei*.

## Discussion

*T. brucei*, *T. cruzi* and *L. major* belong to the Trypanosomatidae family that is evolutionarily distant from the most commonly studied eukaryotes (*H. sapiens*, mouse, *Drosophila melanogaster*, *Caenorhabitits elegans* and *Saccharomyces cerevisiae*) [1,11], representing useful eukaryotic models to explore the evolution of cellular molecular processes. In fact, most of our knowledge of the eukaryotic selenocysteine pathway comes from studies of the mammalian machinery [14]. In this paper, we focused on the trypanosomatid selenophosphate synthetase, a key enzyme in the selenocysteine pathway that is responsible for catalyzing the formation of the biological form of selenium, selenophosphate, for selenocysteine biosynthesis [14,22,38,39]. We determined the crystal structure of an N-terminally truncated selenophosphate synthetase from *L. major* (ΔN-*Lm*SEPHS2) consisting of an AIRS-like fold also conserved in *E. coli* [39], *A. aeolicus* [38,42] and *H. sapiens* [22] orthologs. This result shows that the selenophosphate synthetase fold is an important determinant for its function throughout domains of life.

We further showed that the trypanosomatid selenophosphate synthetase is active as a dimer in solution. Our dimeric model of *Lm*SEPHS2 contains two equivalent unbound active sites in

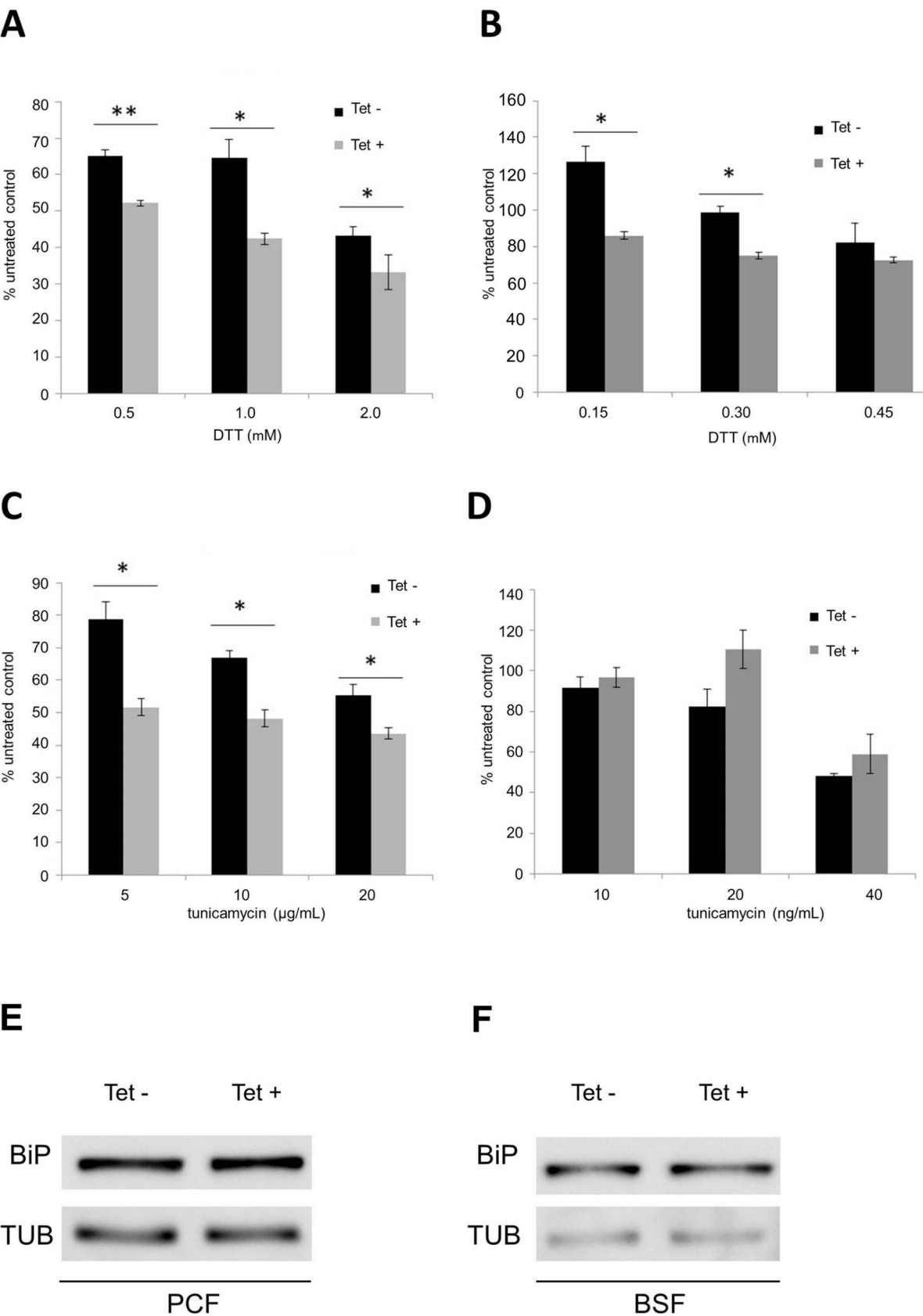

**Fig 5. *Tb*SEPHS2-RNAi *T. brucei* cells sensitivity to DTT and tunicamycin.** Tetracycline non-induced (dark bars) and induced (grey bars) *Tb*SEPHS2 RNAi PCF and BSF *T. brucei* cells treated with various concentrations of DTT and tunicamycin. The plots show cell concentration relative to untreated control after incubation at 28˚C and 37˚C for procyclic (PCF) and bloodstream (BSF) *T. brucei*, respectively. Bars represent the average of 3 independent experiments including standard deviations of experiments proceeded in **A-** and **C-** PCF *T. brucei* cells, and **B-** and **D-** BSF *T. brucei* cells. The asterisks represent significant differences between the stressors of ER treatment (PCF *T. brucei*, 0.5 mM DTT: ** P = 0.007; 1.0 mM DTT: * P = 0.020; 2.0 mM DTT: P = 0.040; 5µg/mL tunicamycin: * P = 0.033; 10µg/mL tunicamycin: * P = 0.010; 20µg/mL tunicamycin: * P = 0.020; BSF *T. brucei*, 0.15 mM DTT: * P = 0.020; 0.3 mM DTT: * P = 0.010; two-tailed Student's t test). Western blot analysis of BiP in whole cell extracts of **I-** PCF and **J-** BSF *Tb*SEPHS2 RNAi *T. brucei* cell (12% SDS-PAGE; α-tubulin as a normalization standard).

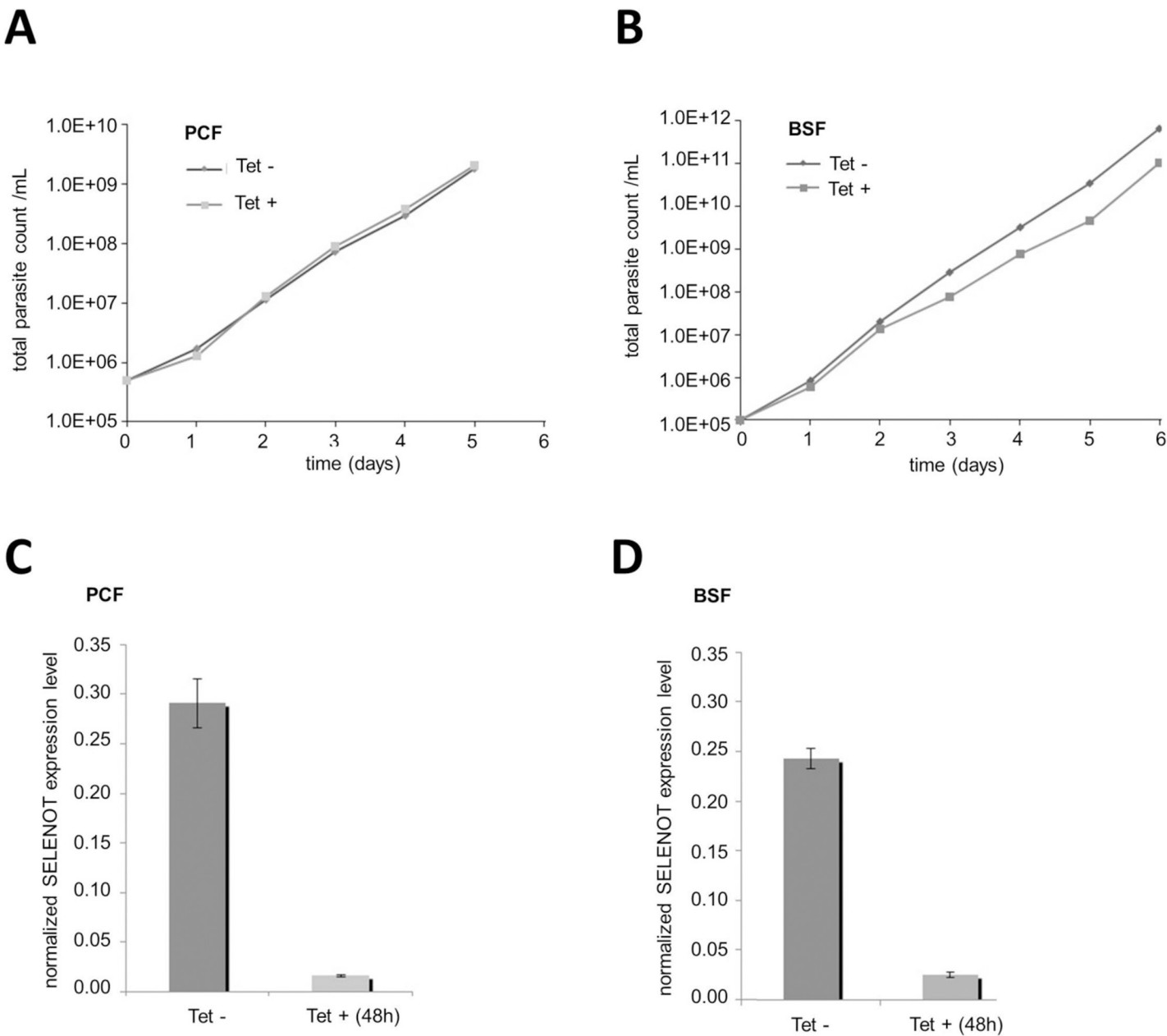

**Fig 6. *Tb*SELENOT is dispensable for both procyclic and bloodstream *T. brucei*.** Growth curves of representative *Tb*SELENOT-RNAi procyclic (PCF) and bloodstream (BSF) *T. brucei* cultures. **A-** and **C-** PCF and **B-** and **D-** BSF *T. brucei* cells induced (black) and non-induced (grey) with tetracycline and real-time qPCR analysis relative to TERT as a normalization standard, respectively.

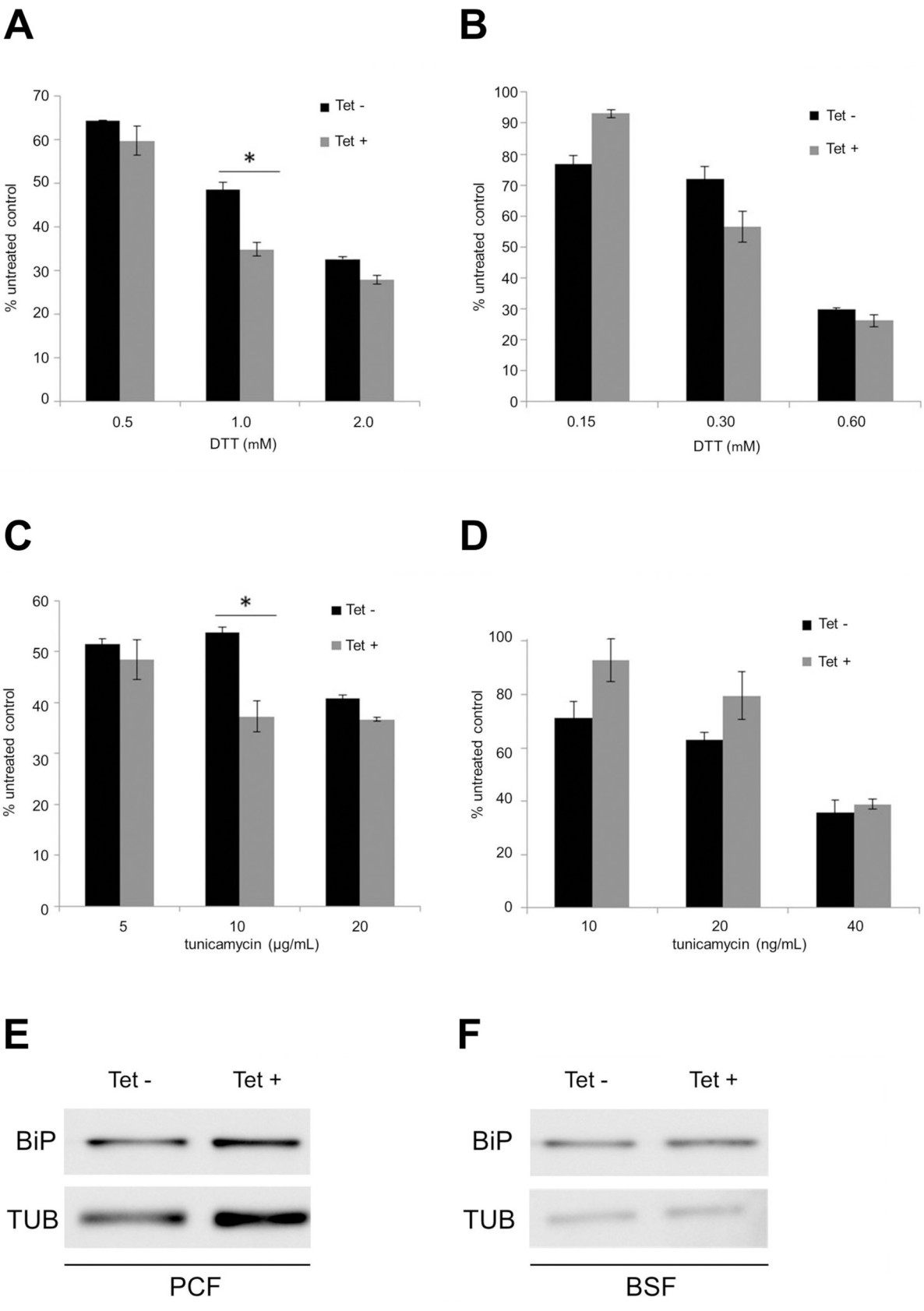

**Fig 7. *Tb*SELENOT-RNAi *T. brucei* cells sensitivity to DTT and tunicamycin.** Tetracycline non-induced (black) and induced (grey) *T. brucei* cells were treated with various concentrations of DTT and tunicamycin. The plots show cell concentration relative to untreated control after a 24h-incubation at 28˚C and 37˚C for procyclic (PCF) and bloodstream (BSF) *T. brucei*, respectively. Bars represent the average of 3 independent experiments including standard deviations of experiments proceeded in **A**- and **C**- PCF *T. brucei* cells, and **B**- and **D**- BSF *T. brucei* cells. Asterisks: PCF *T. brucei*, 1.0 mM DTT: * P = 0.010; 10μg/mL tunicamycin: * P = 0.033; two-tailed Student's t test. Western blot analysis of BiP in whole cell extracts of **E**- PCF and **F**- BSF *Tb*SPS2 RNAi *T. brucei* cells (12% SDS-PAGE; α-tubulin as a normalization standard).

an open conformation prior to ATP and metal binding, similar to what is observed in the apo *E. coli* [39] and *A. aeolicus* [38,42] SEPHS structures. Although partially disordered, the *Lm*SEPHS2 ATP-binding site contains conserved basic amino acid residues that lie in an extended pocket formed between the two amino acid chains of the heterodimer, explaining the need for dimerization of the full-length protein for ATPase activity. A comparison between the apo selenophosphate synthetase crystal structures (*Ec*SEPHS [39] and *Aa*SEPHS [38,42] and *Lm*SEPHS2) and the substrate-bound ones (AMPCPP-*Aa*SEPHS [38], ADP-*Hs*SEPHS1 [22] and AMPcP-*Hs*SEPHS1[22]) suggests that the N-terminal region of the protein, where the catalytic residues are conserved, becomes more ordered upon substrate binding. While the N-terminus of *Ec*SEPHS [39] was observed far from the open ATP-binding site, our crystal structure lacks such a flexible N-terminal region but keeps ATP-binding residues in a similar position. On the other hand, *A. aeolicus* [38] and *H. sapiens* [22] substrate-bound structures of SEPHS showed that the N-terminal region forms a long molecular tunnel suggested to preserve putative cytotoxic Se-containing intermediates from the cytoplasm.

We showed that the crystallized construct (ΔN-*Lm*SEPHS2) does not complement selenoprotein biosynthesis in SEPHS deficient *E. coli* WL400(DE3) strain, as expected due to the lack of catalytic residues. However, a residual ATPase activity was measured *in vitro* in the absence of selenide. This curious result is likely due to the preservation of the ATP-binding site being in the truncated construct. N-terminally truncated constructs of *Tb*SEPHS2 corroborate the data obtained for *Lm*SePHS2. In addition, a comparison between ΔN(25)-*Tb*SEPHS2 and ΔN(70)-*Tb*SEPHS2 functional complementation assays and ATPase activities argues that its first 25 amino acid residues are not essential for selenophosphate synthetase activity in the

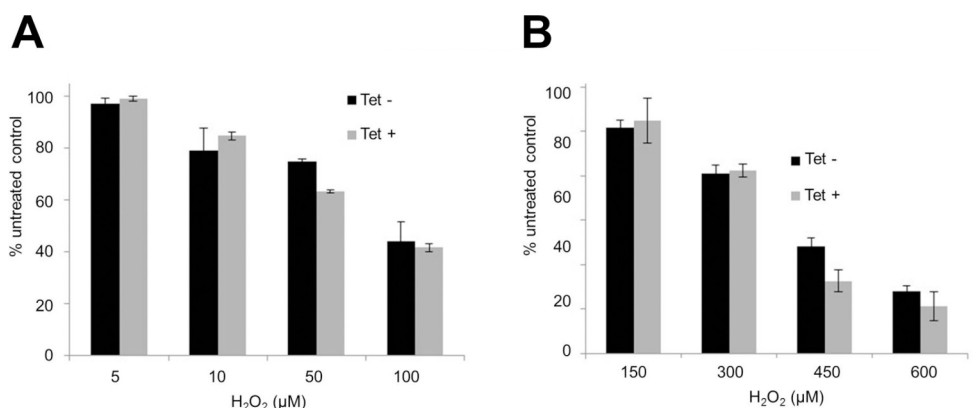

**Fig 8. *Tb*SELENOT knockdown does not affect procyclic and bloodstream *T. brucei* sensitivity to $H_2O_2$.** Tetracycline non-induced (black) and induced (grey) *T. brucei* cells were treated with various concentrations of DTT and tunicamycin. The plots show cell concentration relative to untreated control after a 24h-incubation at 28˚C and 37˚C for procyclic (PCF) and bloodstream (BSF) *T. brucei*, respectively. *Tb*SELENOT-RNAi **A**- PCF and **B**- BSF *T. brucei* cells were also treated with various concentrations of $H_2O_2$. Bars show the cell concentration relative to untreated controls after an 18h-incubation at 28˚C and 37˚C for PCF and BSF *T. brucei*, respectively. The average of 3 independent experiments is shown together with the respective standard deviation.

selenocysteine pathway, but do interfere with ATPase activity. Together, our crystallographic and functional data suggest that ATP-binding is not dependent on the N-terminal region of the protein, although ATPase activity is affected by the presence of such a region.

Interestingly, we previously reported that the disordered N-terminus of *E. coli* SEPHS is involved in the physical interaction between selenophosphate synthetase and selenocysteine synthase and is necessary for selenoprotein biosynthesis [41]. Besides, Itoh et al [38] suggested that the flexibility of the N-terminal Gly-rich loop in selenophosphate synthetase and the per-selenide-carrying loop of selenocysteine lyase (SCLY) would allow the direct transfer of sele-nide between them. The eukaryotic selenocysteine lyase is a pyridoxal 5′-phosphate (PLP)-dependent enzyme that catalyzes the decomposition of selenocysteine into L-alanine and ele-mental selenium [19,20,45]. In fact, the homologous *E. coli* NifS-like proteins support *in vitro* selenophosphate synthesis by SEPHS in the presence of selenocysteine [46]. However, their physical interaction had not been previously established. We demonstrate by SEC-MALS, ITC and fluorescence spectroscopy that *Tb*SEPHS2 indeed binds to *Tb*SCLY *in vitro* in the absence of tRNA[Ser]Sec. The SCLY active site containing PLP is obstructed in the binary complex. We further showed that both *Tb*SEPHS2 and *Tb*SCLY co-purify with each other from *T. brucei* procyclic cells and we observed that such interaction is dependent on the *Tb*SEPHS2 N-termi-nal region. Together, our data suggest that both trypanosomatid selenoprotein biosynthesis and selenocysteine recycling are coupled through the SCLY-SEPHS2 direct interaction (Fig 4A), likely warranting the efficient usage of biological selenium in the cell.

Furthermore, Oudouhou et al [47] also demonstrated that human SEPHS1 and SEPHS2 bind transiently to selenocysteine synthase (SEPSECS) *in vivo*. We observed that PTP-tagged selenocysteine synthase localize to the cytoplasm of PCF *T. brucei*. Interestingly, *Tb*PSTK-PTP co-purified *Tb*SEPSECS and tRNA[Ser]Sec, demonstrating the formation of a stable ternary complex between them. However, *Tb*SEPSECS-PTP did not co-purify with any molecule, indi-cating that its C-terminal PTP-tag might present a steric hindrance for its interaction with *Tb*PSTK. Furthermore, neither *Tb*SEPHS2 nor *Tb*SCLY were observed in other stable com-plexes involved in selenocysteine biosynthesis and incorporation in selenoproteins (Fig 4A), although a report described that selenophosphate synthetase homologs are present in higher order complexes, involved in the selenoprotein biosynthesis pathway in humans [44]. There-fore, our data do not exclude the formation of higher order transient complexes in the Sec pathway in trypanosomatids.

In addition, we showed that 80S ribosomes and polysomes involved in the synthesis of sele-noproteins in PCF *T. brucei* contain *Tb*eEFSec that is dissociated in the presence of a chelating agent. *Tb*eEFSec was also detected in the ribosome-free form. These data suggest that *Tb*eEF-Sec interaction with the ribosome is either dependent on selenoprotein mRNA or take place after 80S ribosome assembly. On the other hand, *Tb*SCLY-SEPHS2 was not detected as associ-ated to ribosomes. Similarly, *Tb*PSTK and *Tb*SEPSECS are mostly detected as ribosome-free complexes. Taken together, our biochemical data indicate that trypanosomatid selenocysteine biosynthesis occurs in a hierarchical process via coordinate action of protein complexes. We hypothesize that, after tRNA[Ser]Sec aminoacylation by SerRS, Ser-tRNA[Ser]Sec may be either specifically transferred to a PSTK-SEPSECS binary complex or to PSTK alone, which phos-phorylates Ser-tRNA[Ser]Sec and subsequently associates with SEPSECS to form a stable com-plex. In archaea, PSTK distinguishes the characteristic D-arm of tRNA[Ser]Sec over tRNA[Ser] [48,49] while human SEPSECS specifically recognizes its 3′-CCA end, TΨC and the variable arm [50]. Hence, Ser-tRNA[Ser]Sec is discriminated from Ser-tRNA[Ser] and mischarged tRNA[-Ser]Sec is avoided. Furthermore, selenophosphate is a toxic compound that is directly delivered to selenocysteine synthase by selenophosphate synthetase via a transient interaction [38,41]. In fact, *Tb*SEPSECS was not copurified with *Tb*SEPHS2 under the conditions tested. In contrast,

a stable *Tb*SEPHS2-*Tb*SCLY binary complex was obtained. Since selenocysteine synthase (SEPSECS) is not specific to selenium compounds [38,51,52], selenophosphate synthetase-selenocysteine lyase complex formation represents another level of fidelity in UGA$_{Sec}$ codon recoding.

Curiously, selenophosphate synthetase has been described as non-essential for *T. brucei* viability [11]. SEPSECS knockout experiments further established that there is no significant contribution of selenoproteins to redox homeostasis in trypanosomatids [11,13,27,28]. Indeed, we further show that *Tb*SELENOT knockdown does not significantly impact PCF and BSF *T. brucei* viability. On the other hand, our group previously demonstrated that *Tb*SEPHS2 is important for oxidative stress response in PCF and BSF *T. brucei* [29]. Accumulation of reactive oxygen species, a common characteristic of oxidative stress, can induce apoptotic cell death in *T. brucei* [53,54]. In addition, the formation of disulfide bonds in ER proteins requires oxidizing power, which has been related to ER oxidoreductin-1 (Ero1) in mammals [55,56,57], a conserved but poorly studied protein in trypanosomatids [58]. Interestingly, protein disulfide isomerase (PDI) and thioredoxin mRNA levels increase in PCF *T. brucei* due to higher mRNA stability under DTT treatment [59]. DTT is thought to interfere with disulfide bond formation leading to accumulation of misfolded proteins in the ER [59]. Besides DTT, chemical ER stress is also commonly achieved with tunicamycin (TN), known to negatively affect N-glycosylation in the ER of *T. brucei* [60]. Here, we demonstrated that ER stress with DTT and TN upon *Tb*SEPHS2 ablation leads to growth defects in PCF *T. brucei*, while only DTT showed a negative effect in BSF *T. brucei* cells expressing *Tb*SEPHS2 RNAi. Together, our data indicate a role for *Tb*SEPHS2 in the ER redox stress response.

Maintaining ER redox is also important for $Ca^{2+}$-dependent cell signaling and homeostasis, which is itself key for mitochondrial homeostasis [61]. In mammals, the ER-resident selenoproteins S, N, K, M and T are believed to regulate the ER redox state, ER stress responses and $Ca^{2+}$ signaling [62]. Among those, SELENOK and SELENOT are conserved in trypanosomatids [12,63]. Interestingly, mammalian SELENOT is localized to the ER and seems to have a thioredoxin-like activity to maintain ER redox homeostasis [64]. SELENOT knockout led to early rat embryonic lethality and its knockdown in corticotrope cells promoted ER stress and unfolded protein response (UPR) [64]. The TrypTAG database [65] contains subcellular localization data for a GFP-tagged *Tb*SELENOT as reticulated in the cytoplasm of the procyclic *T. brucei*, which is consistent with an ER localization. It will be interesting to evaluate its co-localization with ER markers. *Tb*SELENOT-RNAi PCF, but not BSF *T. brucei* cells were only marginally sensitive to DTT and TN at intermediate concentrations. We also verified that both *Tb*SELENOT-RNAi PCF and BSF *T. brucei* cells were not sensitive to increasing levels of hydrogen peroxide. The absence of a strong negative effect of ER stressors in SELENOT-RNAi PCF and BSF *T. brucei* is intriguing. In mammals, SELENOT was identified as necessary for efficient processing of glycosylphosphatidylinositol (GPI) anchoring of proteins [64]. The ER function is particularly necessary for efficient production of GPI-anchored procyclins that coat PCF *T. brucei* cells and of variant surface glycoprotein (VSG) proteins that protects BSF *T. brucei* surface from effectors of the host immune system [66, 67]. However, little is known about the exact role of the ER in *T. brucei* developmental differentiation.

Apart from SELENOT, mammalian SELENOK has been associated with ER homeostasis by promoting calcium flux [68]. However, we were unable to generate a stable SELENOK-RNAi *T. brucei* cell line. It will be informative to evaluate its subcellular localization and function in *T. brucei* since our results from ER stress in selenophosphate synthetase knockdown PCF and BSF *T. brucei* indicate a putative function for the selenocysteine biosynthesis pathway in *T. brucei* ER stress. On the other hand, it is also possible that the only selenophosphate synthetase

isoform observed in trypanosomatids (*Tb*SPSH2) has a more direct role in redox homeostasis in the cell as suggested for the mammalian SEPHS1 [69].

Moreover, we did not observe any alteration in BiP expression upon *Tb*SEPHS2 or *Tb*SELENOT ablation in PCF and BSF *T. brucei*. The presence of UPR is debated in *T. brucei* [32–36, 70]. Goldshmidt et al [59] proposed that a UPR-like pathway is triggered by chemical ER stress in trypanosomatids to reduce the load of proteins to be translocated and enhance degradation of misfolded proteins. Lack of BiP up-regulation upon chemical ER stress in *T. brucei* and *L. donovani* was also observed by Koumandou et al [71], Izquierdo et al [72], Tiengwe et al [32] and Abhishek et al [73], arguing that a UPR-like response based on BiP is inactive in trypanosomatids. On the other hand, these parasites also conserve PKR-like endoplasmic reticulum kinase (PERK) [73], a protein that regulates protein translation by phosphorylating eIF2a, in another mechanism of UPR response in mammals [74]. It is not expected that BSF *T. brucei* could compensate for correct folding of VSGs in the absence of a UPR-like mechanism [66]. New experiments are required to more fully address the molecular response to chemical ER stressors in *T. brucei*.

Although the exact role of the selenocysteine pathway in trypanosomatids is still not clear, we provide new insights into this machinery in these protist parasites. The highly conserved structure of selenophosphate synthetase is essential for selenoprotein biosynthesis across domains of life. Although selenophosphate is not essential for the viability of *T. brucei* [11] and *L. donovani* [13] under laboratory-controlled conditions [11], and some selenoproteins may not be essential as well, as we have shown for selenoprotein T, our data show a role for the sele-nophosphate synthetase in the ER redox stress protection in both the insect stage (procyclic) and the clinically relevant stage (bloodstream) of *T. brucei*. This result is consistent with the global effect of SEPHS2 on the synthesis of selenocysteine and therefore the translation of all selenoproteins. Furthermore, our data stress the need for further investigation of the exact molecular processes involved in *T. brucei* developmental differentiation and upon redox stress, and pH and temperature variation observed in their hosts.

## Materials and methods

### Amino-acid sequence analysis

Amino acid sequences of selenophosphate homologs were retrieved from NCBI [75]: *Aquifex aeolicus* (Aa, WP_010880640.1), *Escherichia coli* (Ec, KPO98227.1), *Pseudomonas savastanoi* (Ps, EFW86617.1), *Phytophthora infestans* (Pi, EEY58478.1), *Trypanosoma cruzi* (Tc, PBJ75389.1), *Trypanosoma brucei* (Tb, EAN78336.1), *Leishmania major* (Lm, XP_00168 7128.1), *Drosophila melanogaster* (Dm_1, AAB88790.1; Dm_2, NP_477478.4), *Mus musculus* (Mm_1, AAH66037.1; Mm_2, AAC53024.2), and *Homo sapiens* (Hs_1, AAH00941.1; Hs_2, AAC50958.2). Amino sequence alignment was generated using Clustal Omega [76].

### Cloning

*Tb*SEPHS2 (Tb927.10.9410), *Lm*SEPHS2 (LmjF.36.5410) and ΔN(69)-*Lm*SEPHS2 cloning was reported previously [36]. ΔN(25)-*Tb*SEPHS2, amino acid residues 26–393, and ΔN(70)-*Tb*SEPHS2, residues 71–393, were cloned into the pET20b expression vector (Novagen) using the following pairs of oligonucleotides: 5'-AGCATATGGGTCTACCGGAAGAGTTTACCT TAACTGAC-3' and 5′- AGCTCGAGAATAATCTTATCATTTACCTTCGCTCCCACCTC-3′, and 5′-AGCATATGGATTGCAGCATTGTGAAACTGCAG-3′and 5′-AGCTCGAGAATA ATCTTATCATTTACCTTCGCTCCCACCTC-3′, respectively. *Tb*SCLY (Tb927.9.12930) was cloned into the pET32a expression vector (Novagen) using the following pairs of oligonucleo-tides: 5'-GGATCCATGTGTAGCATTGAGGGCCCG-3' and 5'-CTCGAGTTACTAAAACT

CACCGAACTGTTGC-3'. For the PTP (Protein A-TEV site-Protein C) tagged protein constructs, the ORFs were amplified using the following primers that contained *Apa*I and *Eag*I restriction sites: *Tb*SEPHS2 5'-GGGCCCGTCTCAAATGATCCGTCCAACAG-3' and 5'-C GGCCGAATAATCTTATCATTTACCTTC-3', *Tbe*EFSec (Tb927.4.1820) 5'-GGGCCCCATC ACGTTTGAATGCCCTTC-3' and 5'-CGGCCGCTGCTGAAGCTGACTGTGGAG-3', *Tb*SEP-SECS (Tb927.11.13070) 5'-GGGCCCGCCGCCATTCGACTGGGTCGTG-3' and 5'-CGGCC GTACCCCCTCGACCGGCCAAAC-3', *Tb*PSTK (Tb927.10.9290) 5'-CGGCCGATGACA GTTTGTCTTGTTCTAC-3' and 5'-GGGCCCCTCGCCAAACACTTCGACTTC-3', *Tb*SCLY 5'-GGGCCCCTATTGATGACCTCGTGAAAC-3' and 5'-CGGCCGAAACTCACCGAACTG TTGCAC-3'. Constructs were designed for homologous expression of C-terminally PTP-tagged protein, with exception of *Tb*PSTK, which was cloned into PN-PTP plasmid. Prior to transfection, the constructs were linearized with the *Bsm*I, *Afl*II, *Nsi*I, *Xcm*I and *Xcm*I restriction enzymes, respectively. *Tb*SEPHS2-RNAi silencing was carried out with the construct described by Costa et al. [29] and *Tb*SELENOT-RNAi was achieved with a specific fragment PCR-amplified from PCF *T. brucei* genomic DNA using gene-specific primers 5'- CCGATTTGTTCGCATCTC ATTTTC-3' and 5'- ACCAGAGATAATTTGGCGCAG -3' and cloned into a modified p2T7$^{TA-blue}$ with phleomycin resistance.

## Recombinant protein purification

*Tb*SEPHS2, ΔN(25)-*Tb*SEPHS2, ΔN(70)-*Tb*SEPHS2, *Lm*SEPHS2 and *ΔN-Lm*SEPHS2 were expressed in *E. coli* and purified as described previously [36]. *Tb*SCLY (Tb927.9.12930) expression was induced by IPTG in *E. coli* BL21(DE3) for 16 hours at 18˚C. Cells were harvested and sonicated in lysis buffer (50 mM HEPES pH 7.5, 300 mM NaCl, 2% glycerol, 10 mM imidazole, 5 mM DTT, 1X cOmplete Protease Inhibitor Cocktail (Roche)) supplemented with 10 μM PLP. The lysate was centrifuged at 40,000 X g for 20 min at 4˚C and the supernatant was applied to a 5 ml Ni-NTA Superflow Cartridge (Qiagen) in ÄKTA Purifier 10 (GE). The product was dialyzed against the same buffer in the absence of imidazole, incubated with 1 mM PLP on ice and subsequently washed 5 times with the same buffer. The product was applied to a Superdex 200 10/300 column (GE) and concentrated using an Amicon ultracentrifugal filter.

## *In vitro* activity assay

Recombinant selenophosphate synthetase constructs at 30 μM final concentration were added independently to a reaction mixture (100 μl) containing 50 mM Tris-HCl pH 7.5, 50 mM KCl, 5 mM MgCl2, 5 mM DTT, 1 mM ATP and incubated at 26˚C. The reaction was blocked by incubation at 75˚C for 10 minutes and the solution was centrifuged at 16,000 x g for 45 minutes. The nucleotides present in the supernatant were separated by high-pressure liquid chromatography (HPLC) using a Waters Alliance 2695 HPLC with a C18 reversed-phase column (5 μm, 15 cm x 4.6 mm inside diameter, SUPELCOSIL LC-18-S; Sigma Aldrich) equipped with a guard column at a flow rate of 1 ml/min. The mobile phase consisted of buffer A (50 mM potassium phosphate buffer ($KH_2PO_4$/$K_2HPO_4$), pH 4.0) and a gradient of buffer B (20% methanol vol/vol in buffer A). The gradient conditions were: 5 min—100% buffer A at 1.0 ml/min; 10 min—100% buffer A at 0.1 ml/min; 1 min—100% buffer B at 0.5 ml/min; 1 min—100% buffer B at 1.0 ml/min; 1 min 100% buffer B at 1.0 ml/min. Nucleotide peaks were detected, and the peak area for ATP (maximum absorbance at 254 nm) was calculated relative to the respective control (enzyme absence). Raw data are available in S2 Table.

## Functional complementation assay in *Escherichia coli*

Functional complementation experiments were conducted according to Sculaccio et al [26] for N-terminally truncated selenophosphate synthetase constructs. Briefly, *E. coli* WL400(DE3) was transformed with the full-length and truncated constructs of both *T. brucei* and *L. major* SEPHS2 and the cells were analyzed for the presence of active selenoprotein formate dehydrogenase H (FDH H) using the benzyl viologen assay under anaerobic conditions at 30˚C for 48 hours.

## Native gel electrophoresis

Recombinant *Tb*SEPHS2, ΔN(25)-*Tb*SEPHS2, ΔN(70)-*Tb*SEPHS2, *Lm*SEPHS2 and *ΔN-Lm*SEPHS2 were applied to a PhastGel gradient 8–25% (GE) at 1 mg/mL at room temperature.

## Analytical ultracentrifugation (AUC)

*Tb*SEPHS2 and *Lm*SEPSH2 at 0.15, 0.30, 0.45, 0.60 and 0.80 mg/mL in 25 mM Tris pH 8.0, 50 mM NaCl, 1mM β-mercaptoethanol were subjected to velocity sedimentation at 30,000 rpm at 20˚C in an An60Ti rotor using a OptimaTM XL-A analytical ultracentrifuge (Beckmann). The data were analyzed with SEDFIT [77] using a c(s) distribution model. The partial-specific volumes (v-bar), solvent density and viscosity were calculated using SEDNTERP (Dr Thomas Laue, University of New Hampshire). Raw data are available in S2 Table. To determine the tetramer-dimer equilibrium dissociation, 110 μL of protein at concentrations of 0.5, 0.75 and 1.0 mg/mL were loaded in 12 mm 6-sector cells and centrifuged at 8,000, 10,000 and 12,000 rpm at 4˚C until equilibrium had been reached. Data were processed and analyzed using SEDPHAT [77].

## Size exclusion chromatography with multi angle light scattering (SEC-MALS)

The molecular mass distribution of *Tb*SCLY (40 μM), *Tb*SEPHS2 (40 μM) and *Tb*SCLY-*Tb*SEPHS2 (40 μM:40 μM) in solution was determined using SEC-MALS. A 1:1 *Tb*SCLY-*Tb*SEPHS2 mixture was incubated in 50 mM HEPES pH 7.5, 300 mM NaCl, 2% glycerol, 10 mM imidazole, 5 mM DTT at 25˚C for 30 minutes and 60 minutes on ice previous to the SEC-MALS analysis. 40 μl of each sample was loaded onto a WTC-030N5 (Wyatt) column running at 0.3 ml/min coupled to a mini-DAWN TREOS equipped with an Optilab rEX detector (Wyatt). Data was analyzed using Astra 7.0.1.24 (Wyatt). Raw data are available in S2 Table. Experiments were performed at room temperature.

## Isothermal titration calorimetry (ITC)

ITC measurements were performed at 25˚C in a VP-ITC calorimeter (Microcal) using 50 mM Tris-HCl, pH 7.4, 100 mM NaCl, 1 mM DTT buffer. Samples of *Tb*SCLY at 10 nM in the calorimeter cell were titrated with *Tb*SEPHS2 at 200 μM from the syringe using 29 injections of 2 μL preceded by a 0.5 μL injection. The resulting excess heats associated with the injections were integrated and normalized using the measured concentrations of protein from UV absorbance, corrected for the background heat of dilution of *Tb*SEPHS2 and the binding data were fitted to a two-species hetero-association model using Microcal ORIGIN software (Microcal). Raw data are available in S2 Table. We were unable to perform the potentially informative reverse titration, with *Tb*SCLY in the syringe, because of its limited stability at high concentration at 25˚C.

## Fluorescence spectroscopy

Fluorescence spectroscopy of the pyridoxal phosphate (PLP) group bound to *TbSCLY* excited at 450 nm was performed in an ISS-PC spectrofluorometer (ISS). Varying concentrations of *Tb*SEPHS2 (1:0.5, 1:0.75, 1:1, 1:1.5) were incubated in 50 mM HEPES pH 7.5, 300 mM NaCl, 2% glycerol, 10 mM imidazole, 5 mM DTT with 20 µM *Tb*SCLY at 25˚C for 30 minutes and 60 minutes on ice. Fluorescence spectroscopy was then performed at room temperature. Raw data are available in S2 Table.

## Circular dichroism (CD) spectroscopy

Purified *Tb*SEPHS2, *Lm*SEPHS2 and their truncated constructs were dialyzed against 25 mM Tris pH 8.0, 50 mM NaCl, 1mM β-mercaptoethanol at 4˚C overnight and adjusted to a concentration of 0.2 mg/mL. Purified *Tb*SCLY at 0.2 mg/mL was kept in the same buffer. Far-UV CD spectra at 5˚C were measured using a Jasco J-815 spectropolarimeter (JASCO) (0,1 nm resolution, 100 nm/min, quartz 0.5 cm cuvette).

## X-ray crystallography

X-ray diffraction data collection and analysis were previously reported for both *Tb*SEPHS2 and ΔN-*Lm*SEPHS2 crystals. Although the data was not sufficient for *Tb*SEPHS2 structure determination, ΔN-*Lm*SEPHS2 final model was completed by molecular replacement with PHASER [78] using *H. sapiens* SEPHS1 [22] structure coordinates as search model (PDB code 3FD5). Model building and refinement were performed using PHENIX [79] and COOT [80]. Structure visualization was performed in PyMOL. ConSurf [81] was used for conservation analysis.

## RNAi experiments

*Trypanosoma brucei* procyclic RNAi 29–13 cell line (derived from strain 427) was grown in Cunningham's culture media [82] supplemented with 10% tetracyclin-free fetal bovine serum (FBS) (Atlanta Biologicals) in the presence of G418 (15 µg/mL) and hygromycin (50 µg/mL) to maintain the integrated genes for T7 RNA polymerase and tetracycline repressor, respectively. *T. brucei* bloodstream form 221 (Variant Antigen Type expressing VSG, from strain 427), was cultured in HMI-9 media supplemented with 10% tetracycline-free FBS, supplemented with G418 (15 µg/mL).

The *Tb*SELENOT RNAi construct was obtained using a fragment between the positions 9 to 754 according to Costa et al. 2011. Stable *Tb*SEPHS2 RNAi cell lines were transfected using the constructs described by Costa et al. 2011. Clonal populations of transfected cells were obtained by limiting dilutions and selected with 2.5 µg/mL of phleomycin (Sigma). To monitor the growth of RNAi cells, dsRNA synthesis was induced with 1 µg/mL of tetracycline. The cells were counted daily and diluted to a concentration of 2 x$10^5$ cells/mL.

For bloodstream *T. brucei* transfection, we used the protocol described by Burkard et al. 2011 [83]. In summary, 10 µg of *Not*I-linearized DNA were used per 6×$10^7$ cells in 100µl homemade Tb-BSF buffer (90mM sodium phosphate, 5mM potassium chloride, 0.15mM calcium chloride, 50mM HEPES, pH 7.3). Electroporation was performed using 2mm gap cuvettes (BTX, Harvard apparatus) with program X-001 of the Amaxa Nucleofector (Lonza). Following each transfection, stable transformants were selected for 6 days with 2.5 µg/mL pheomycin as a pool culture in 125 ml HMI-9 medium containing 10% tetracycline-free FBS.

Mid-log phase *T. brucei* procyclic form (5 x $10^5$ cells/mL) and *T. brucei* bloodstream form (1 x $10^5$ cells/mL) were treated with tetracycline for 48 and 24 hours, respectively, aimed to

induce the RNAi. These cells were incubated with various DTT and tunicamycin concentrations for 4 hours or $H_2O_2$ for 8 hours and cell viability was confirmed by staining with fluorescein diacetate. Raw data are available in S2 Table.

## Real time RT-PCR (qPCR)

RNA from 1 to $2x10^7$ trypanosomes bloodstream and procyclic forms, respectively, were isolated with NucleoSpin RNA II Kit, (Macherey-Nagel), and 1.2 μg of RNA were reverse transcribed by SuperScript III Reverse Transcriptase (Life Technologies). To quantify levels of specific mRNA transcripts in individual samples, 1 μl of each cDNA sample was amplified with gene-specific primers in iQ SYBR Green Supermix (Bio-Rad) according to the manufacturer's protocol, using a C1000 thermocycler fitted with a CFX96 real-time system (Bio-Rad Laboratories). Three technical and three biological replicates of each reaction were performed. Every test gene was normalized to the mRNAs encoding TERT, which is known to be expressed constitutively [84]. The temperature profile was: 95˚C, 3min [(95˚C, 15 s; 55.5˚C, 1 min; data collection) 40˚C].

## Polysomal profile analysis

For polysome profile analysis, 500 mL of each PTP-TAP *T. brucei* culture were grown overnight at 28˚C to mid-log phase and cells were treated with cyclohexamide 100 μg/mL for 5 minutes. The cultures were immediately chilled on ice and collected by centrifugation at 3,000 X g at 4˚C for 7 minutes. Cells were washed twice with ice-cold Salts buffer (Tris-HCl 10 mM pH 7,5; KCl 30 mM; $MgCl_2$; 10 mM; DTT 1 mM; cyclohexamide 100 μg/mL), pellet volume was estimated and suspended with the same volume of Lysis buffer (Tris-HCl 10 mM pH 7,5; KCl 30 mM; $MgCl_2$; 10 mM; DTT 1 mM; cyclohexamide 100 μg/mL; 1,2% Triton), supplemented with 1X Protease cOmplete inhibitor cocktail (Roche). Lysates were clarified by centrifugation at 17,000 X g for 15 minutes. Eight hundred micrograms equivalent of $OD_{260}$ units was loaded on a 7–47% sucrose gradient prepared in Salts buffer and centrifuged at 39,000 RPM for 2 hours at 4˚C in a Beckman SW41Ti rotor. The gradients were fractionated by upward displacement with 60% (w/v) sucrose using a gradient fractionator ISCO UA-6 UV Vis with Type 11 optical unit at 254 nm and fractions were collected manually for subsequent western blotting analysis. Subunit profile analysis was performed as described before with modifications; lysis buffer was supplemented with 50 mM of EDTA and lysed was centrifuged on a 5–25% sucrose gradient for 4 hours.

## PTP-tag TAP and mass spectrometry

PTP-tagged protein expression was demonstrated by immunoblotting with a polyclonal rabbit anti-protein A antibody (SIGMA) as described previously [13]. Tandem affinity purification (TAP) was performed according to a standard PTP purification protocol [42]. Briefly, 4.5 L of procyclic PTP-transfected *T. brucei* culture was grown until mid-log phase, which corresponded to approximately 5–7 x $10^6$ cells/mL. The cell pellet obtained by centrifugation at 3.000 X g was washed twice with cold Tryps wash buffer (20mM Tris-HCl pH 7,5, 100mM NaCl, 3mM $MgCl_2$), and once with cold Transcription buffer (20 mM HEPES/KOH pH 7.7, 150 mM sucrose, 20 mM potassium L-glutamate, 3 mM $MgCl_2$), and its volume was estimated and re-suspended with the same volume of cold Transcription buffer supplemented with 1 mM DTT and a tablet of protease inhibitor cocktail mini cOmplete (Roche). The cells were lysed in a FRENCH Press system (ThermoFisher scientific) at 20,000 psi, alternating 1 min under pressure and 1 min on ice, for 6 cycles. The lysate was mixed with 1/10 v/v Extraction buffer (20 mM TRIS/HCl pH 7.7, 150 mM KCl, 3 mM $MgCl_2$, 0.5 mM DTT and 1% v/v

Tween20), incubated for 20 min on ice, centrifuged at 21,000 X g, 4˚C and the soluble fraction was submitted to IgG Sepharose 6 Fast Flow chromatography purification (GE), using resin previous equilibrated with PA-150 buffer (20 mM TRIS/HCl pH 7.7, 150 mM KCl, 3 mM MgCl$_2$, 0.5 mM DTT and 0.1% v/v Tween 20). After washes, the tagged proteins were eluted with TEV protease (300 U, Promega) in 4 mL PC-150 buffer (20 mM TRIS/HCl pH 7.7, 150 mM KCl, 3 mM MgCl$_2$, 1 mM CaCl$_2$, 0.5 mM DTT and 0.1% v/v Tween 20) supplemented with cOmplete (Roche) protease inhibitor cocktail. A second chromatography step was carried out in an Anti-Protein C Affinity Matrix (Roche) pre-equilibrated with PC-150 buffer and co-purified proteins were recovered with Elution buffer (5 mM TRIS/HCl pH 7.7, 10 mM EGTA, 5 mM EDTA and 0.01 mg/mL leupeptin). 15μL of StrataClean resin (Stratagene) was added to 1.8 mL of eluate, centrifuged at 3,000 X g and the sample was separated in 12% SDS-PAGE stained with SYPRO Ruby (Invitrogen).

For protein identification, bands were excised from the gel, de-stained with 50% methanol and 5% acetic acid for 5 minutes, dehydrated with 100% acetonitrile for another 5 minutes and reduced with 5 mM DTT for 30 min at room temperature, following alkylation with 14 mM of iodoacetamide in the dark for 30 min. Protein digests were carried out with 0.75 μg of trypsin (SIGMA) for 16 h at 4˚C at 900 rpm, and reactions were stopped with 5% formic acid and bands dried in vacuum. Digestion products were desalted using ZipTipC18 (Merck) according to the manufacturer's instructions. Peptides were suspended in 0.1% formic acid and injected in an in-house made 5 cm reversed phase pre-column (inner diameter 100 μm, filled with a 10 μm C18 Jupiter resins -Phenomenex) coupled to a nano-HPLC (NanoLC-1DPlus, Proxeon) online to an LTQ-Orbitrap Velos (ThermoFisher Scientific). The peptide fractionation was carried out in an in-house 10 cm reversed phase capillary emitter column (inner diameter 75 μm, filled with 5 μm C18 Aqua resins-Phenomenex) with a gradient of 2–35% of acetonitrile in 0.1% formic acid for 52 min followed by a gradient of 35–95% for 5 min at a flow rate of 300 ml/min. The mass spectrometry was operated in a data-dependent acquisition mode at 1.9 kV and 200˚C. MS/MS spectra were acquired at normalized collision energy of 35% with FT scans from m/z 200 to 2000 and mass resolution of 3 kDa. Raw data were processed in Proteome Discovery 1.3 using MASCOT as a search engine and the complete database of *T. brucei* obtained from TriTrypDB [32].

## RNA analysis

Reverse transcription (RT)-PCR experiments were used to monitor the presence of tRNA[Ser][Sec] in selenocysteine protein complexes. 100 μl cell extracts were incubated with 30 μl of IgG sepharose 6 fast flow (GE) beads, equilibrated with PA-150 buffer. After five washes, total RNA was extracted by TRIzol reagent (GE) and the first strand synthesized by SuperScript III reverse transcriptase (Invitrogen) with random hexamers primers (ThermoFisher scientific). PCR was performed with tRNA[Ser][Sec] sense (5'-GCGCCACGATGAGCTCAGCTG-3') and tRNA[Ser][Sec] antisense (5'-CACCACAAAGGCCGAATCGAAC-3') oligonucleotides.

## Fluorescence microscopy

*T. brucei* cell lines expressing PTP-tagged proteins were used for immunolocalization assays. Briefly, $5 \times 10^6$ cells were washed with PBS buffer pH 7.4 (SIGMA), fixed with 2% v/v paraformaldehyde for 20 min at 4˚C, permeabilized with 0.3% v/v Triton X-100 for 3 min at 4˚C. Cells were blocked with 3% w/v BSA and incubated with 1:16,000 v/v rabbit anti-protein A antibody (SIGMA) for 1 h at room temperature. After washes, cells were incubated with 1:400 (v/v) Alexa Fluor 594-conjugated (Invitrogen) and 10 μg/mL DAPI (SIGMA) at room temperature. The coverslips were mounted on glass slides with Vectashield mounting medium

(Vector Laboratories) and images obtained with Olympus IX-71 (Olympus) inverted microscope coupled Photometrix CoolSnapHQ CCD camera were further deconvoluted using DeltaVision (Applied Precision) software.

## Supporting information

**S1 Fig. Multiple amino acid sequence alignment.** Conserved residues are highlighted (catalytic Cys/Sec in yellow and Lys in green, and ATP binding amino acids in red). Secondary structures are also shown (arrow: α-helix; rectangle: β-strand). Amino acid sequences: Lm–*L. major* (XP_001687128.1), Tb–*T. brucei* (EAN78336.1), Tc–*T. cruzi* (PBJ75389.1), Pi–*Phytophthora infestans* (EEY58478.1), Dm—*Drosophila melanogaster* (Dm_1: AAB88790.1, Dm_2: NP_477478.4), Mm—*Mus musculus* (Mm_1: AAH66037.1, Mm_2: AAC53024.2), Hs–*Homo sapiens* (Hs_1: AAH00941.1, Hs_2: AAC50958.2), Ps—*Pseudomonas savastanoi* (EFW86617.1), Aa–*Aquifex aeolicus* (WP_010880640.1), Ec—*Escherichia coli* (KPO98227.1).
(TIF)

**S2 Fig. Electrophoresis. A**- Coomassie blue-stained native gel electrophoresis of *T. brucei* and *L. major* selenophosphate synthetase constructs: 1- *Tb*SEPHS2, 2- ΔN(70)-*Tb*SEPHS2, 3- ΔN(25)-*Tb*SEPHS2, 4- *Lm*SEPHS2, and 5- ΔN(69)-*Lm*SEPHS2. Major bands correspond to dimers. The second most abundant species in each lane corresponds to the respective tetramer. **B**- Coomassie blue-stained SDS-PAGE of *T. brucei* selenocysteine lyase.
(TIF)

**S3 Fig. Sedimentation equilibrium analytical ultracentrifugation (SE-AUC).** Multi-speed and multi-concentration global fitting for a dimer-tetramer self-association system of **A**- *Tb*SEPHS2 ($K_d = 161 \pm 10$ μM) and **B**- *Lm*SEPHS2 ($K_d = 178 \pm 10$ μM). Fitting residuals are shown as insets.
(TIF)

**S4 Fig. Circular dichroism (CD) spectroscopy.** CD spectra for **A**- selenophosphate synthetase constructs (*Tb*SEPHS2, ΔN(25)-*Tb*SEPHS2, ΔN(70)-*Tb*SEPHS2, *Lm*SEPHS2 and ΔN-*Lm*SEPHS2, and **B**- *T. brucei* selenocysteine lyase (*Tb*SCLY).
(TIF)

**S5 Fig. ITC. A**- ITC data for SCLY-buffer (circle), buffer-SEPHS2 (triangle) and SEPHS2-SCLY (star), and **B**- SCLY-ΔN(70)-SEPHS2 titration experiments using VP-ITC calorimeter and analyzed in NITPIC.
(TIF)

**S6 Fig. Selenocysteine pathway machinery localization in procyclic *T. brucei* cells.** PTP-tagged proteins immunolocalized using anti-protein A antibody (red). DAPI (blue) is used as a nuclear/kinetoplast marker. Untransfected *procyclic T. brucei* 427 cells were used as negative controls.
(TIF)

**S1 Table. Mass spectrometry identification of co-eluted proteins with selenocysteine machinery components.**
(XLSX)

**S2 Table. Raw data.**
(XLSX)

## Acknowledgments

We would like to acknowledge Dr. Susana Andrea Sculaccio Beozzo from the São Carlos Physics Institute, University of São Paulo for technical support, Dr. Ana Carolina Migliorini Figueira from the Laboratory of Spectroscopy and Calorimetry of the National Laboratory of Biosciences (LNBIO, Brazil) for AUC, and Dr. Andressa Patricia Alves Pinto from the São Carlos Physics Institute, University of São Paulo for SEC-MALS and ITC. The anti-BiP antibody was kindly provided by James D. Bangs, School of Medicine and Biomedical Sciences, State University of New York at Buffalo and the anti-EIF5A antibody was gently supplied by Sergio Schenkman, Departament of Microbiology, Immunology and Parasitology Federal University of São Paulo.

## Author Contributions

**Conceptualization:** Marco Túlio Alves da Silva, Ivan Rosa e Silva, Otavio Henrique Thiemann.

**Investigation:** Marco Túlio Alves da Silva, Ivan Rosa e Silva, Lívia Maria Faim, Natália Karla Bellini, Murilo Leão Pereira, Ana Laura Lima, Teresa Cristina Leandro de Jesus, Fernanda Cristina Costa, Tatiana Faria Watanabe, Bidyottam Mittra, Norma W. Andrews, Otavio Henrique Thiemann.

**Methodology:** Marco Túlio Alves da Silva, Ivan Rosa e Silva, Humberto D'Muniz Pereira, Júlio Cesar Borges, Marcio Vinicius Bertacine Dias, Júlia Pinheiro Chagas da Cunha.

**Resources:** Otavio Henrique Thiemann.

**Supervision:** Sandro Roberto Valentini, Cleslei Fernando Zanelli, Bidyottam Mittra, Norma W. Andrews, Otavio Henrique Thiemann.

**Writing – original draft:** Marco Túlio Alves da Silva, Ivan Rosa e Silva, Otavio Henrique Thiemann.

**Writing – review & editing:** Otavio Henrique Thiemann.

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
