## [Decision Letter · Decision Letter 0]

29 Apr 2020

Dear Dr Thiemann,

Thank you very much for submitting your manuscript "Trypanosomatid selenophosphate synthetase structure, function and interaction with selenocysteine lyase" for consideration at PLOS Neglected Tropical Diseases. As with all papers reviewed by the journal, your manuscript was reviewed by members of the editorial board and by several independent reviewers. In light of the reviews (below this email), we would like to invite the resubmission of a significantly-revised version that takes into account the reviewers' comments. The reviewers raised major concerns regarding the presentation and interpretation of results and the Discussion section.

We cannot make any decision about publication until we have seen the revised manuscript and your response to the reviewers' comments. Your revised manuscript is also likely to be sent to reviewers for further evaluation.

Sincerely,

Igor C. Almeida

Associate Editor

Hans-Peter Fuehrer

Deputy Editor

Reviewer's Responses to Questions

**Key Review Criteria Required for Acceptance?**

**Methods**

-Are the objectives of the study clearly articulated with a clear testable hypothesis stated?

-Is the study design appropriate to address the stated objectives?

-Is the population clearly described and appropriate for the hypothesis being tested?

-Is the sample size sufficient to ensure adequate power to address the hypothesis being tested?

-Were correct statistical analysis used to support conclusions?

-Are there concerns about ethical or regulatory requirements being met?

Reviewer #1: Methodology is generally sound.

For minor issues, see section 'Editorial and Data Presentation Modifications'.

Reviewer #2: The objectives and methodological design of the study are clear and suited to adress the first. 

Some concerns are outlined in the sections below.

Reviewer #3: (No Response)

**Results**

-Does the analysis presented match the analysis plan?

-Are the results clearly and completely presented?

-Are the figures (Tables, Images) of sufficient quality for clarity?

Reviewer #1: The analysis matches the analysis plan. The results are in general clear and completely presented, and the figures are clear besides some minor issues.

For detailed, minor issues, see section 'Editorial and Data Presentation Modifications'.

Reviewer #2: The results match the analysis plan however some results are not properly presented (see comments below). Figures and images are well presented (see minor comments in Section Data Presentation Modifications).

- the truncated form of SEPHS2 was cryztallized as a monomer, however, in solution, the protein appears to display a dimeric arrangement. May the author comment whether the N-terminal portion of SEPHS2 contributes to protein dimerization?, another question, is wether the disordered N-terminal region of SEPHS2 may contribute to an anomalous behavior of the protein in solution; May the autors add to the Table from Fig. 1G the theoretical sedimentation coefficient expected for a SEPHS2 monomer?

- please indicate in the legend of Fig 3C the concentration of each protein species used for the titration as well as the corresponding mass centers of the spectrum for each condition tested. protein 

- please indicate whether single clones or whole population of SEPHS2- and SELENOT-RNAi T. bruce were subjected to phenotypic analysis. 

- please check the following text because the references to Figs appears to be missquoted "Interestingly, SELENOT-RNAi-induced PCF cells were more sensitive to TN

than DTT (Figures 6F and 6H). On the other hand, sensitivity to different concentrations of DTT

varied in SELENOT-RNAi- BSF T. brucei, with induced cells being more sensitive to 350-400 uM

DTT. No significant effect was observed in the presence of TN (Figure 6E-H)." BSF T. brucei depleted in SELENOT did not show any significant sensitivity towards DTT or TN, when compared to non-induced parasites. 

- for the following sentence: "Treatment with different concentrations of hydrogen peroxide did not affect T. brucei growth (Figure 6J-K),", please highlight that such conclusion is valid for both parasite life stages: PCF and BSF.

Reviewer #3: (No Response)

**Conclusions**

-Are the conclusions supported by the data presented?

-Are the limitations of analysis clearly described?

-Do the authors discuss how these data can be helpful to advance our understanding of the topic under study?

-Is public health relevance addressed?

Reviewer #1: The conclusions drawn from the data are in general warranted. The Discussion is detailed, appropriate and places the result in the biological context. 

For detailed, minor issues, see section 'Editorial and Data Presentation Modifications'.

Reviewer #2: Most conclusions are correct, except for some related to the biological results obtained with different RNAi cell lines. For instance, some conclusions related to the sensitivity of the RNAi cell lines towards DTT or tunicamycin are somehow (and likely unvoluntary) missleading. 

The limitations of the present study and the prospective for new assays demanded to solve some questions are well identified by the authors.

For instance, in the Synopsis and in the Conclusions it is stressed the concept that "Our results also show how the interaction of different proteins leads to the protection of the cell against the toxic effects of seleium compounds during selenocysteine synthesis". I disagree wit this statement since depletion of SEPHS, which should lead to an accumulation of toxic selenium, proved not to entail any detrimental effect on procyclic and bloodstream T. brucei (Ref. 11 and 27). Even for both parasite stages of T. brucei, the KO of SEPSECS, which should lead to an accumulation of toxic selenium metabolites, did not render a defective phenotype in vitro and in vivo (Ref. 11, 27, 28). Instead, may it be possible that the SEPHS2/SCLY complex warrants an efficient recycling of the trace element Selenium?

- Please revise the following statement "Here, we demonstrated that ER stress

with DTT and TN upon TbSEPHS2 ablation leads to growth defects in both PCF and BSF T. brucei,

indicating a role for TbSEPHS2 in the ER stress response.", because the experimental data presented for BSF and TN do not support such conclusion.

- Please revise the following statement "Indeed, we observed that SELENOT knockdown PCF and BSF T. brucei cells were also sensitive to ER stress using DTT,...". This is not true because data presented in Fig. 6 shows that BSF depleted in SELENOT are fully insensitive to TN and DTT treatment, and PCF showed a marginal sensitivity towards these stressors at single (out of three) concentrations tested. Thus, the behavior was not concentration-dependent and thus, probably irrelevant. If the authors agree with that, then, they should reconsider not to stress, along the text, a role for SELENOT in ER redox homeostasis, as their data are not fully conclusive in this respect. 

- There is not direct evidence from the biological data produced in this study to support the following statement "Our data suggests that ER N-glycosylation of proteins is strictly regulated in trypanosomatids." First, because non of the protein studied in this work was shown to be localized at the ER; second, because the impact on N-glycosylation of downregulating Sel-metabolism in T. brucei has not been addressed in this study.

Reviewer #3: (No Response)

**Editorial and Data Presentation Modifications?**

Reviewer #1: A general comment: Next time, please number pages and/or lines to make the work of reviewers a bit easier.

Scientific issues:

1. Abstract, line 1: The description ‘early-branching eukaryotes’ is outdated since it is known, since about two decades, that they did not branch early from a common eukaryotic phylogenetic trunk as previously thought. The current view is that different superphyla originated already at the beginning of the eukaryotic evolution (LECA) and trypanosomatids belong to a different superphylum (Excavata) than vertebrates/fungi (Opisthokonta).

2. Abstract, line 17: Should the sentence not state ‘only SELENOT-RNAi procyclic T. brucei cells were sensitive’ (as opposed to bloodstream cells)? Nonetheless, I am not convinced by the conclusion, because the effect of DTT between induced and non-induced cells seems only significant for one DTT concentration used, not for the other two (see also related comments below).

3. Introduction, end of 4th paragraph: Mention if the TbSEPHS2 RNAi experiments were done with both PCF and BSF cells.

4. Introduction, 6th paragraph, line13: Same as above: I am not convinced by the conclusion, because the effect of DTT between induced and non-induced cells seems only significant for one DTT concentration used.

5. Results, Section 1 (The L. major selenophosphate synthetase ..), end of paragraph 2: What is the basis for the conclusion that ‘Our crystal structure suggests that the conserved AIRS-like fold of selenophosphate synthetases is necessary for their enzymatic mechanism.’? Or did the authors meant to conclude: ‘Our finding that the AIRS-like fold of selenophosphate synthetases is also conserved in our Leishmania crystal structure suggests its necessity for the enzymatic mechanism.’? 

6. Results, Section 1, last sentence: I suppose a sulfate ion was identified in the crystal structure. Could it be highlighted in the cartoon representation (with its mentioning in the legend)? Moreover, it seems relevant to add the information also in the legend of Fig. S1.

7. Results Section 4 (TbSEPHS2 RNAi-induced …..) and Fig. 5: The observation that low concentrations (0.15-0.30 mM) of DTT affect growth in TbSEPHS2 RNAi-induced BSF cells, but a higher concentration (0.45 mM) not (and all concentrations do in PCF cells) is intriguing and may deserve further discussion.

8. Results Section 5 (Selenoprotein T (SELENOT)): The second part of this section is not easy to follow. I have the impression that, both in the text and the legend, the labelling of panels E – H has been mixed up. My understanding is that panels E and G are about PCF cells and F and G about BSF cells (= similar organization of panels as in Figure 5). 

In that case, the text should probably changed as follows:

line 11: ‘(Figures 6E and 6G)’ to ‘(Figures 6E – 6H)’

line 12: ‘(Figures 6F and 6H)’ to ‘(Figures 6E and 6G)’

line 14: add ‘(Figure 6E-H)’ after ‘DTT’

line 14: ‘(Figure 6E-H)’ to ‘(Figure 6F and 6H)’

and the following change in the legend, line 8: 

‘E- and F- PCF T. brucei cells’ to ‘E- and G- PCF T. brucei cells’

‘G- and H- BSF T. brucei cells’ to ‘F- and H- BSF T. brucei cells’

Furthermore, I am puzzled by two issues in this section: 

(1) The conclusion mentioned in the Abstract and the end of the Introduction about the sensitivity of RNAi-ablated PCF cells for DTT (although only significant at one concentration) is surprisingly not addressed in this section.

(2) I don’t see how to conclude from Fig. 6 (panel F?) that ‘On the other hand, sensitivity to different concentrations of DTT varied in SELENOT-RNAi- BSF T. brucei, with induced cells being more sensitive to 350-400 uM’. First, there seem no significant difference between induced and non-induced cells. Second, the figure mentioned 0.60 mM, not 0.40 mM.

9. Discussion: The sentence ‘Here, we demonstrated that ER stress with DTT and TN upon TbSEPHS2 ablation leads to growth defects in both PCF and BSF T. brucei, indicating a role for TbSEPHS2 in the ER stress response.’ is not entirely accurate. As demonstrated in Fig. 5D (and correctly mentioned elsewhere in the manuscript), TN does not have such effect in BSF trypanosomes.

10. Discussion: About the sentence ‘Indeed, we observed that SELENOT knockdown PCF and BSF T. brucei cells were also sensitive to ER stress using DTT, ….’, see my comments above (for PCF cells only significant at one concentration (Fig 6E), for BSF cells results seem not convincing at all (Fig. 6F)).

11. Figure 1: in the various cartoons a loop is highlighted with X. Please explain in the legend.

Minor comments on the biological nomenclature used:

1. Synopsis, lines 3 and 6 and Introduction line 1: Replace the old, incorrect word ‘protozoa’ by ‘protists’ (‘parasitic protists’ or ‘protist parasites’).

2. Introduction, line 1 of 5th paragraph: Change old taxonomic name ‘Kinetoplastida’ to the correct ‘Kinetoplastea’.

3. Introduction, 6th paragraph, lines 1 and 2: no upper case E for the (English) word eukaryotes.

4. M&M, RNAi experiments: ‘host strain 29-13’. 29-13 is not a strain but a cell line derived from strain 427.

5. M&M, RNAi experiments: ‘strain 221’. 221 is not a strain but a VAT (Variant Antigen Type expressing VSG no 221) from strain 427.

Minor comments on the presentation:

1. Abstract, line 2: Add s to parasite to give parasites.

2. Abstract, lines 12-13: Sentence not very clear. Is meant: ‘….. phosphoseryltRNA Sec kinase (PSTK)-Sec-tRNASec synthase (SEPSECS) complex and the tRNASec-specific elongation factor (eEFSec)-ribosome complex’? If so add ‘complex’ (2x).

3. Abstract line 13: add h to ‘ditiothreitol’ to give ‘dithiothreitol’

4. Introduction, end 2nd paragraph: Change ‘a detailed …. network and … mechanism … remain poorly understood’ to ‘details of …. remain poorly understood’.

5. Introduction, 5th paragraph, line 12: change ‘result’ to ‘results’.

6. Introduction, 5th paragraph, line 17: add ‘of’ to give ‘decrease of mitochondrial’

7. Introduction, 6th paragraph, line5: Add ‘The to give ‘TbSEPHS2 crystal structure ….’

8. Results, Section 1 (The L. major selenophosphate synthetase ..), line 6: Change A to Å.

9. Results, Section 1, line 27: Fig. 1H is not found.

10. Results, Section 3 (T. brucei SEPHS2 binds …), line 8: add, after ITC: ‘(not shown)’

11. Results, Section 3, line 12: Why ‘Remarkably’ which suggests some surprise? Would ‘Importantly’, or ‘Noteworthy’ not be better?

12. Results, Section 3,lines 26 and 27: Are (Figure 3E) in line 26 and (Figure 3D) in line 27 not swapped?

13. Results, Section 3, line 37: Change ‘present mammalian in’ to ‘present in mammalian’

14. Results, Section 3, line 39: (Figure 4C) should be (Figure 4D), whereas in line 41 (Figure 4D) should be (Figure 4E)

15. Legend Fig 1, line 9: Correct ‘sedmentation’ to ‘sedimentation’

16. Legend Fig 1, line 11: Change ‘insert’ to ‘inset’. 

17. Legend Fig 2, line 4: Correct ‘defficient’ to ‘deficient’

18. Legend Fig 4, line 3: Correct ‘riobosome’ to ‘ribosome’

19. Legend Fig 6, line 13: Change 0 in H202 to O

20. M&M, at several places: ‘xx mM de HEPES’. What is de meaning of ‘de’? (see sections: Recombinant protein purification, Size exclusion chromatography, Fluorescence spectroscopy)

21. M&M, RNAi experiments: SM-9? Should this not be SM-79 (the medium described for PCF T. brucei by Brun & Schonenberger, Acta Trop, 1979).

22. M&M, RNAi experiments, line 10: Change ‘it was used the protocol described’ to ‘the protocol used has been described’

23. M&M, RNAi experiments, line 11: Change ‘sum’ to ‘summary’

24. M&M, RNAi experiments, line 3 from end: Delete d from ‘induced’

25. M&M, PTP-tag TAP, line 7: Change ‘digest’ to its plural form

26. Legend Fig. S1: Mention the indications for secondary structures

27. Legend Fig. S3, line 4: Change ‘insert’ to ‘inset’

Reviewer #2: Rephrase the following sentences, all of which are confusing:

- "TbSEPHS2 crystal structure we determined demonstrates that a conserved fold is important for its function."

- 

Figure 1D is not quoted in the main text. It may removed, or, instead, the B-factors can be shown in the structure shown in Fig1A.

- Fig. S5 should be quoted in the following sentence: "Additionally, we observed that ΔN(70)-TbSEPHS2 does not bind TbSCLY as measured by ITC, indicating that the N-terminal region is necessary for in vitro interaction,..."

- In the following sentence, "... for TbeEFSec-PTP (Figure 3E). RT-PCR analysis revealed that tRNA[Ser]Sec co-precipitates with the TbPSTK-PSEPSECS complex and with TbeEFSec (Figure 3D).", swap the quotation to Fig3E and Fig3D, and in the legend of Fig3E, please explain what does input mean. 

- In teh following sentence "Besides, they are representatives of the Trypanosomatidae

family that is evolutionarily distant from the most...", the word "representative" may be replaced by "belong to".

- "SELNOT" should be replaced by "SELENOT" in the following sentence "Interestingly, SELNOT knockout led to early rat embryonic lethality and its knockdown in corticotrope"

Reviewer #3: (No Response)

**Summary and General Comments**

Reviewer #1: The manuscript by Thiemann and coworkers provides a lot of new information about proteins involved in selenocysteine biosynthesis in Leishmania and Trypanosoma, especially about the selenophosphate synthetase of these parasitic protists. The work builds upon previous research by this group. The information is novel and interesting. The research has been well performed and the results are discussed in great detail and placed in context. The manuscript has been well written. Nonetheless, I have a number of comments, questions and suggestions concerning the scientific content, and identified a number of minor issues about the presentation.

Reviewer #2: The study by da Silva et al. addresses structural, biochemical and biological aspects of several components of the selenocysteine metabolism of African trypanosomes. Following the sucessfull crystallization of the selenophosphate synthase, they solve and describe the 3D structure of an N-terminally truncated for of this protein. They investigated the quaternary arrangement of the protein by means of ultracentrifugation and gel filtration approaches, and showed in vitro and in cell the formation of an heterocomplex between the selenophosphate synthase, with selenocystein synthase. Using heterologous-complementation assays, they also demonstrate the residues and regions functionally relevant for selenophosphate synthase activity. Finally, they addressed a potential role of the selenophosphate synthase and the selenocysteine-containing protein SELENOT in the response to reductive stress.

Overall, the methodological strategy and results are consistent with most of the conclusions elaborated. At some point the authors should make clear why this metabolism deserves interest, taking into account that it has been shown not to be indispensable for the in vitro and in vivo survival of the clinically relevant stage of African trypanosomes (as well as for the insect stage). Is there any possibility that this metabolism may play a role during parasite differentiation? 

At several part of the Result and Discussion sections there is redundancy in the presentation and discussion of the results. I would recommend that the authors make an effort to limit themselves to simply describe and interpret the results without major framework discussion.

Reviewer #3: (No Response)

PLOS authors have the option to publish the peer review history of their article (what does this mean?). If published, this will include your full peer review and any attached files.

Reviewer #1: Yes: Paul Michels

Reviewer #2: No

Reviewer #3: No
---

## [Decision Letter · Decision Letter 1]

3 Aug 2020

Dear Dr Thiemann,

We are pleased to inform you that your manuscript 'Trypanosomatid selenophosphate synthetase structure, function and interaction with selenocysteine lyase' has been provisionally accepted for publication in PLOS Neglected Tropical Diseases.

Best regards,

Igor C. Almeida

Associate Editor

Hans-Peter Fuehrer

Deputy Editor

Reviewer's Responses to Questions

**Key Review Criteria Required for Acceptance?**

**Methods**

-Are the objectives of the study clearly articulated with a clear testable hypothesis stated?

-Is the study design appropriate to address the stated objectives?

-Is the population clearly described and appropriate for the hypothesis being tested?

-Is the sample size sufficient to ensure adequate power to address the hypothesis being tested?

-Were correct statistical analysis used to support conclusions?

-Are there concerns about ethical or regulatory requirements being met?

Reviewer #1: OK

Reviewer #2: (No Response)

Reviewer #3: (No Response)

**Results**

-Does the analysis presented match the analysis plan?

-Are the results clearly and completely presented?

-Are the figures (Tables, Images) of sufficient quality for clarity?

Reviewer #1: OK

Reviewer #2: (No Response)

Reviewer #3: (No Response)

**Conclusions**

-Are the conclusions supported by the data presented?

-Are the limitations of analysis clearly described?

-Do the authors discuss how these data can be helpful to advance our understanding of the topic under study?

-Is public health relevance addressed?

Reviewer #1: OK

Reviewer #2: (No Response)

Reviewer #3: (No Response)

**Editorial and Data Presentation Modifications?**

Reviewer #1: OK

Reviewer #2: (No Response)

Reviewer #3: (No Response)

**Summary and General Comments**

Reviewer #1: The authors have appropriately dealt with all my comments on the original version of their manuscript.

Reviewer #2: The authors proceedded with the requested changes and answered satisfactory all the questions raised in my revision.

There are still present a few typing mistakes that can be identified and corrected upon a carefull reading by the authors.

Reviewer #3: Overall, I was satisfied with the answers to my comments, and in my opinion, the quality of the manuscript has also increased significantly after incorporating all the recommendations of other reviewers. In conclusion, the manuscript meets all the criteria for publishing in PLOS Neglected Tropical Diseases.

PLOS authors have the option to publish the peer review history of their article (what does this mean?). If published, this will include your full peer review and any attached files.

Reviewer #1: **Yes: **Paul Michels

Reviewer #2: **Yes: **Marcelo A. Comini

Reviewer #3: No

---

## [Editor Report · Acceptance letter]

29 Sep 2020

Dear Dr Thiemann,

We are delighted to inform you that your manuscript, "Trypanosomatid selenophosphate synthetase structure, function and interaction with selenocysteine lyase," has been formally accepted for publication in PLOS Neglected Tropical Diseases.

Best regards,

Shaden Kamhawi

co-Editor-in-Chief

Paul Brindley

co-Editor-in-Chief
